# TRUTH OR DECEIT? A BAYESIAN DECODING GAME ENHANCES CONSISTENCY AND RELIABILITY

## ABSTRACT

Large Language Models (LLMs) often produce outputs that – though plausible – can lack consistency and reliability, particularly in ambiguous or complex scenarios. Challenges arise from ensuring that outputs align with both factual correctness and human intent. This is problematic in existing approaches that trade improved consistency for lower accuracy. To mitigate these challenges, we propose a novel game-theoretic approach to enhance consistency and reliability during the decoding stage of LLM output generation. Our method models the decoding process as a multistage Bayesian decoding game. This ensures consistency through *Correctness Alignment* and enhances reliability via *Ambiguity Calibration*. The model dynamically converges to a consensus on the most reliable outputs and distinguishes {Valid, Specious} outputs without human feedback or additional training. Remarkably, our game design allows smaller models to outperform much larger models through game mechanisms (*e.g.* 78.1 LLaMA13B *vs* 76.6 PaLM540B), as well as integrating various LL strategies and models, demonstrating the potential of game-theoretic tools to improve the truthfulness and reliability of LLMs.

## 1 INTRODUCTION

Large Language Models (LLMs) have demonstrated extraordinary capabilities in tasks such as factual question answering, fact-checking, and open-ended text generation (Brown et al., 2020; Radford et al., 2021). However, as these generative models increase in complexity and scale, they not only enhance their generation capabilities but also develop a tendency to produce outputs that, while plausible, may be factually incorrect or subtly misleading (McKenzie et al., 2023). This "specious" behavior – whether an inevitable artifact of the model's optimization process or an unintended hallucination (Banerjee et al., 2024; Bai et al., 2024) – poses a significant challenge, often outpacing the ability of human judgment to accurately assess the fidelity and truthfulness of the generated content (Leike et al., 2018). One direct approach is to optimize model outputs for legibility via human feedback (e.g., RLHF (Christiano et al., 2017; 2018; Saunders et al., 2022; Markov et al., 2023)). Human feedback, inherently constrained by limitations in interpretability (Singh et al., 2024) and hindered by the illegibility (Hendrik Kirchner et al., 2024) of AI-generated content, struggles to keep pace with increasingly complex reasoning (Casper et al., 2023; Leike et al., 2018). In light of these challenges, the reliability of generative models in collaborative and high-stakes decision-making remains deeply uncertain and we pose the question:

> How can we efficiently ensure that LLM outputs are not only aligned with human intent but also **valid**, especially when human evaluation may overlook **specious** errors?

To answer this question and to address the challenges outlined in Fig. 1, we explore a game-theoretic approach by introducing a *Verifier* as a dynamic and rigorous evaluator, serving as a proxy for human judgement to systematically assess generators. The motivation for this approach is threefold: (1) LLMs are increasingly employed to assist in evaluating their own outputs, offering a more scalable alternative to solely relying on human feedback (Bai et al., 2023; Saunders et al., 2022; Markov et al., 2023; Mu et al., 2024); (2) the flexibility to adjust game-theoretic objectives – such as utilities and policies between the generator and verifier – allows us to analyze latent decoding consistency and legibility as a function (Jacob et al., 2024; Hendrik Kirchner et al., 2024); and (3) in scenarios where human guidance is constrained, structured AI interactions can effectively elicit and refine

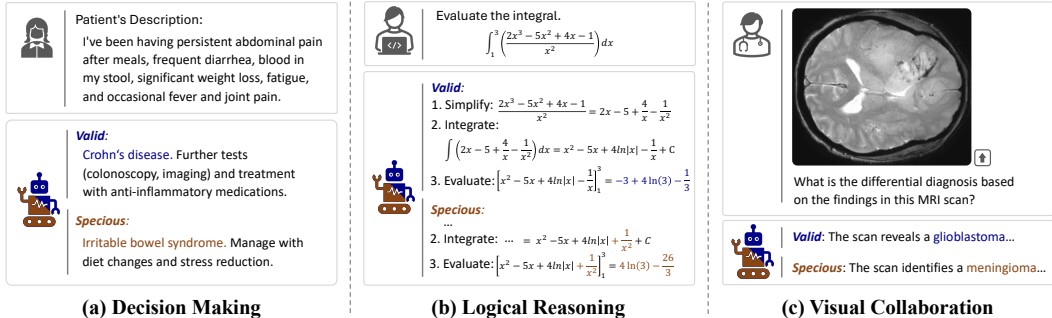

**(a) Decision Making**  **(b) Logical Reasoning**  **(c) Visual Collaboration**

Figure 1: **Distinguishing Valid from Specious LM outputs, particularly when human evaluation may overlook plausible errors.** The three panels demonstrate how models can generate both Valid (accurate and reliable) and Specious (plausible but misleading) responses.

latent knowledge, thereby enhancing model reliability, and generation consistency (Christiano et al., 2021; Turpin et al., 2024).

**Focus of the paper.** Realistically, neither models nor humans can be expected to be perfectly correct or reliable. Thus, our work focuses on actual consistency in *correctness* and calibrated ambiguity in *reliability*. Instead of perfect correctness, we expect the LLM generator to have a high pass rate and the LLM verifier to have a high recall on helpful samples, driving them towards a robust equilibrium of consensus. Instead of absolute reliability, we aim to identify false negative and false positive samples that exhibit fluctuating or unstable behavior under calibrated confidence and disambiguity metrics during the decoding stage, using these elusive cases as focal points for a more precise and targeted verification process. We categorise these cases as

1. **Valid Output from Generator *vs.* Verifier**. When both the generator and verifier agree on correct results and are reliable to both humans and the LLM.

2. **Specious Output from Generator *vs.* Verifier**. When the generator and verifier agree on results that seem correct but are actually wrong. In these cases, the verifier may recognize the result, even though a human might not.

This is analogous to expecting reliable outputs when humans can consistently discern correctness and are not misled by similar-looking but opposite intents. To address the problem, we propose a signaling game setup between a generator and a verifier to resolve the current issues in LLM decoding. This is motivated by concepts from Equilibrium Consensus Games (ECG) (Jacob et al., 2024) and Prover-Verifier Games (PVG) (Hendrik Kirchner et al., 2024). Conventional signaling game settings have been successfully deployed for Poker (Brown & Sandholm, 2018; 2019), Stratego (Perolat et al., 2022), Diplomacy (, FAIR; Bakhtin et al., 2022; Jacob et al., 2022), and LLM tasks (Hendrik Kirchner et al., 2024; Chen et al., 2023).

Variable correctness and ambiguous reliability requires us to formulate this as a multi-step Bayesian Decoding Game with complex action spaces as shown in Fig. 2. Firstly, the generator's design is randomly sampled from {Correct, Incorrect} outputs to match latent alignment and consistency, Then, the verifier judges the type of decoding from generators {Valid, Specious} based on a convex combination of *correctness* and *reliability*. Both the generator and verifier iteratively refine their policies using improved no-regret optimization until they reach an equilibrium of correctness judgement, followed by solving a constrained optimization problem.

## 2 A BAYESIAN DECODING GAME (BDG)

We define generative model decoding as a signaling game (§2.1), where the generator and verifier iteratively exchange signals to align on the correctness of the generated outputs. We introduce a Bayesian Game for Generative Model Decoding (§2.2) to counter potential pitfalls of an unconstrained equilibrium – where multiple equilibria can lead to suboptimal outcomes. We introduce No-regret Optimization for Separating equilibria (§2.3) to minimize cumulative regret and to ensure

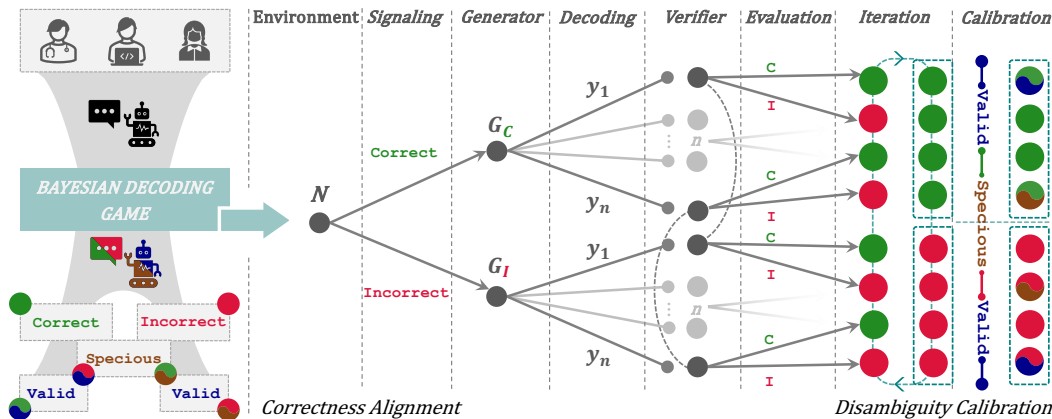

Figure 2: A *Bayesian Decoding Game* **ensures consistency through** *Correctness Alignment* **and enhances reliability via** *Ambiguity Calibration*. The generation and verification are structured as a multi-stage signaling game, fostering a coherent consensus on the outputs {Correct, Incorrect} while improving the reliability {Valid, Specious}.

that strategies converge towards a distinct separation. Finally, to ensure decoding is not only correct but also reliable, we introduce Constrained Optimization for Ambiguity Calibration (§2.4), which refines candidate outputs based on a convex combination of correctness and disambiguity metrics.

## 2.1 PRELIMINARIES: SIGNALING GAME FOR GENERATIVE LANGUAGE MODELS

The essence of a signaling game (Gibbons et al., 1992) is that one player (the generator) takes an action, the signal, to convey information to another player (the verifier); in the simplest setup, the final payoff depends on whether the verifier correctly judges the generator's type based on the generator's signal. Following this intuition, (Jacob et al., 2024) design a Equilibrium Consensus Game (ECG), without a formal definition of the game. Thus, we provide a comprehensive game-theoretic formulation for generative model decoding, and propose improvements to address limitations.

Formally, the signaling game's components can be defined as: (1) *Players*: Generator and Verifier; (2) *Choice sets*: Generator's choice set is $y \in \mathcal{C}_G = \mathcal{Y}$, with prompt $p$ randomly drawn from {Correct, Incorrect}, and the Verifier's choice set is $v \in \mathcal{C}_V = \{$Correct, Incorrect$\}$, based on the generator's choice $y \in \mathcal{Y}$; (3) *Payoff Function*: $u_G = u_V = \mathbb{1}_{p=v}(p, v)$, where $\mathbb{1}$ equals 1 if the correctness prompt $x$ matches the verification result, and 0 otherwise. This basic signaling game is illustrated in Fig.2. We are now ready to state the fundamental concept of this signaling game, a Perfect Bayesian Nash Equilibrium (PBNE) (Cho & Kreps, 1987). We use the short form Perfect Bayesian Equilibrium (PBE). The auxiliary definitions for PBE Defi. 1. are in Appx. C.

**Definition 1.** *(Perfect Bayesian Equilibrium (Fudenberg, 1991)) A Perfect Bayesian Nash Equilibrium (PBE) is a pair $(s, b)$ of strategy profile and a set of beliefs such that (1) $s$ is sequentially rational given beliefs $b$, and (2) $b$ is consistent with $s$.*

**Example.** For generative model decoding, the generator's belief is given by its perceived probability distribution, $\mathbb{P}(\{$**correct**, **incorrect**$\}) = (p_i, 1 - p_i)$, for each $y_i \in \mathcal{Y}$ of the verifier's judgment, and with its belief and type, the generator chooses a mixed action that maximizes its utility, i.e., if the generator's type is **correct**, then its optimal mixed strategy would be allocating positive possibility only on $y_i$ such that $p_i > 1 - p_i$ and zero possibility to other $y_i$.

**Collusion Avoidance with SE (Sequential Equilibrium).** A player in the game is said to be sequentially rational if and only if, at each information set (the nodes of Generator and Verifier in Fig. 2) they are to move, they maximize their expected utility given their beliefs in the information set. Given any strategy profile $s$ and any information set $I$ on the path of play of $s$, a player's beliefs at $I$ are said to be consistent with $s$ if and only if the beliefs are derived using the Bayes' rule and $s$. Collusion occurs when the reached equilibrium is uncorrelated with the generator's type. Thus, we further introduce a Separating Equilibrium (SE) to our game to avoid collusion.

**Definition 2.** *(Separating Equilibrium (Black et al., 2012)) A Separating Equilibrium (SE) is a type of PBE where agents with different types choose different actions.*

To enable the LLM to differentiate between correct and incorrect answers while reaching consensus, we require the generator to respond distinctively to each type. When provided with correct or incorrect inputs, the generator must exhibit different behaviors, sending clear signals to the verifier.

**Theorem 1.** *More than one (mixed) strategy Perfect Bayesian Equilibrium exists for this game.*

The proof and explanation are in Appx. D.1. Thm. 1. is both a guarantee and a curse; the existence of an equilibrium ensures convergence, but the presence of multiple equilibria raises the risk of undesirable outcomes, where low-quality output may incorrectly align with successful verification.

## 2.2 Game Formulation: A Bayesian Game for Generative Model Decoding

The game converges towards a desirable outcome following equilibrium for the problem at hand:

**Definition 3.** *(Decoding Game) The Decoding Game is an alternative version of the signalling game in §2.1, and its payoff is determined by the preference ordering of each player, $O_i \in S_{\mathcal{Y}}, i \in \{G, V\}$, where $n$ is the cardinality of the candidate set $\mathcal{Y}$ and $S_n$ is the set of all permutations of elements in $\mathcal{Y}$. We define the utility of the decoding game as*

$$u_G(O_G, O_V) = u_V(O_G, O_V) = \mathbb{1}_{(O_G = O_V)}(O_G, O_V),$$

such that $\mathbb{1}_{(O_G = O_V)}(\cdot, \cdot)$ is the indicator function at $O_G = O_V$. $O_i$ is the preference relation indicated by players' strategy, $a_G(y \mid x, \text{correct}, b_S), a_V(\text{correct} \mid x, y, b_V)$, such that

$$a_G(y_i \mid x, \text{correct}, b_S) \geq a_G(y_j \mid x, \text{correct}, b_S) \iff y_i \succsim_G y_j$$

$$a_V(\text{correct} \mid x, y_i, b_V) \geq a_V(\text{correct} \mid x, y_j, b_V) \iff y_i \succsim_V y_j.$$

$b_G = b_G(y \mid x, \text{correct})$ is the generator's belief of the probability of $y$ being judged correctly by the verifier, and $b_V = b_V(\text{correct} \mid x, y)$ is the verifier's belief of the probability of $y$ being associated with the correct environment signal received by the generator. $a_G, a_V$ are the actions for the generator and verifier, respectively. [1] With the preference relation, we determine $O_G, O_V$, and we call the equilibrium for the above game a Decoding Equilibrium (DE).

**Theorem 2.** *There are $n!$ equilibria for the Decoding Game.*

The proof and details are given in Appx. D.2. Thm. 2. can guide a refined utility with a penalty term to avoid undesirable equilibria, (*e.g.,* change of preference ordering, collusion). We write this as

$$\begin{aligned}
u_G(a_G, a_V) &= u_V(a_G, a_V) \\
&= \mathbb{1}_{(O_G = O_V)}(O_G, O_V) - \max\{\|a_G(y \mid x, \text{correct}, b_G) - a_{NV}(\text{correct} \mid x, y, b_V)\|, \|\forall y\}
\end{aligned} \tag{1}$$

Utility is maximized when the preferences of the verifier and generator match, and when the largest absolute difference is minimized given a valid signal, which each correspond to (1) a Decoding Game, and (2) $\sigma$-close, consistent with the defi. of $\sigma$-DE in Appx. D.2 for an optimal convergence.

## 2.3 Correctness Alignment: No-regret Optimization for Equilibrium

**No-Regret Optimization.** Based on the Decoding Game (§2.2), we propose two strategy update schedules to numerically achieve optimal convergence of $\sigma$-DE in Thm. 1. The multiplicity of DE may lead to convergence to suboptimal outcomes, necessitating the definition of an initial strategy for each player. This "true" prior is denoted as $a_V^{(1)}(\cdot \mid x, y)$ and $a_G^{(1)}(\cdot \mid x, v)$ (Jacob et al., 2024).

Through repeated interactions and iterative policy refinement, no-regret learning approximates equilibria in large games. Cumulative regret is defined as:

$$\text{Reg}_i^{(T)} := \frac{1}{T}\left(\sum_{t=1}^{T} u_i\left(s_i^*, s_D^{(t)}; b_i\right) - u_i\left(s_i^{(t)}, s_D^{(t)}; b_i\right)\right),$$

---

[1]There is a difference between **belief** and **action**: the **belief** is the player's belief in the opponent's action.

where $s_i^*$ is the optimal hindsight strategy that maximizes this value. Rather than computing regret at each iteration, $s_i^*$ is selected based on the time-averaged strategy profile.

In sequential games with private information and discrete choices, global regret minimization is achieved by minimizing regret locally within each information set, given the finite nature of these sets. For example, to minimize overall regret, the generator must minimize regret by selecting an optimal mixed strategy $s_G$, conditioned on the signal correctness received from the environment. The verifier follows a similar procedure, updating its strategy with respect to each $y \in \mathcal{Y}$.

For this problem the payoff is maximized when the generator and verifier align their actions and minimize their confidence difference. Thus, the strategy update should be directed towards alignment with the opponent's strategy based on the adaptability[2] of players (Roughgarden, 2010).

**Markovian Strategy Update.** To maximize the utility given by Eq. 1, whereas each player's belief $b_{i,t}$ at time $t$ of the opponent's strategy is given by the opponent's strategy in period $t-1$. We hence propose a Markovian strategy update schedule. The palyers update their strategy based on the belief:

$$b_G^{(t+1)}(y \mid x, v) = a_V^{(t)}(v \mid x, y), \quad b_V^{(t+1)}(v \mid x, y) = a_G^{(t)}(y \mid x, v)$$

$$a_G^{(t+1)}(y \mid x, v) \propto \exp\left\{ \frac{\frac{1}{2} b_G^{(t+1)}(y \mid x, v) + \lambda_G \log a_G^{(t)}(y \mid x, v, b_G^{(t)})}{1/(\eta_G t) + \lambda_G} \right\}$$

$$a_V^{(t+1)}(v \mid x, y) \propto \exp\left\{ \frac{\frac{1}{2} b_V^{(t+1)}(v \mid x, y) + \lambda_V \log a_V^{(t)}(v \mid x, y, b_V^{(t)})}{1/(\eta_V t) + \lambda_V} \right\}.$$

Initial policies are $a_V^{(1)}(\cdot \mid x, y)$, $a_G^{(1)}(\cdot \mid x, v)$, where $\eta_i, \lambda_i, i \in \{G, V\}$ are the learning rate and stiffness hyperparameter. The two strategy update schedules we propose show satisfactory convergence properties, and the stopping criteria are given by: (1) $O_G = O_V$. (2) $|a_G(y \mid x, \text{correct}, b_S) - a_{NV}(\text{correct} \mid x, y, b_V)\| < \sigma$.

**Theorem 3.** *A Markovian update schedule for a Decoding Game converges to an equilibrium.*

The proof can be found in Appx.D.3. Due to the efficient utility and the design of the no-regret algorithm, our method reaches $\sigma$-DE 30 times faster than other state-of-the-art methods with an accurate correctness alignment between the generator and verifier. Moreover, the strategy update also ensures that the convergence of $\sigma$-DE is a "near" convex combination of the initialization of the generator and the verifier.

**Inherent Ambiguity.** Once the $\sigma$-DE is reached, given $\mathbf{P}((\{\text{correct, incorrect}\}) = (0.5, 0.5)$ is the signal distribution from the environment, and the cardinality of the candidate set is given by $|\mathcal{Y}| = n$ so that $n \mod 2 = 0$, we label the n most preferred candidates as correct, and the rest as incorrect. We denote the candidate in each group as $y_{i,C}, y_{i,I}$, respectively. However, sometimes in experiments we found that in equilibrium[3], candidates' correctness near the cutoff threshold are actually ambiguous. This is expected, for example, if the correctness score difference between the least correct and the least incorrect candidate $y_{\frac{n}{2},C}, y_{\frac{n}{2}+1,I}$,

$$P_{LM}^*(\text{correct} \mid x, y_{\frac{n}{2},C}) - P_{LM}^*(\text{correct} \mid x, y_{\frac{n}{2}+1,I})$$

is small compared to the difference between other adjacent candidates. Consequently, when running the signaling game with the same set of candidates and prompt, the equilibrium preference may become indeterminate for candidates whose correctness is near the threshold.

Through NLP metrics in (Li et al., 2022) and human evaluation (see Appx. G for details), we observe that they lack both similarity to the prompt and logical progression in natural language. This leads to increased perplexity, resulting in false positive and false negative classifications. Therefore, we refer to these output candidates as **specious** due to the inherent ambiguity in model sampling.

A uniform probability distribution determines the cutoff boundary of natural's signal, we aim to solve this ambiguity problem by introducing an independent (from correctness) metric to detect those **specious** candidates and allow for post-equilibrium checks and decoding optimizaiton.

---

[2]An adaptive player is a function that inputs (1) the opponent $i$, (2) time $t$, (3) mixed strategies $s^1, \dots, s^t$ produced by $i$, and (4) past actions $a^1, \dots, a^{t-1}$, and outputs a coupled strategy and belief.

[3]we denote the equilibrium actions by *

## 2.4 DISAMBUGUITY CALIBRATION: OPTIMIZATION VIA A DISAMBIGUITY METRIC

Our game efficiently approximates the $\sigma$-DE, which is accomplished solely based on the correctness judgment of both the generator and the verifier. While correctness in LLM-generated text is undoubtedly the most important metric, the ease with which this correctness can be verified has been largely overlooked. (Hendrik Kirchner et al., 2024) tapped this problem where they reasoned the property of *Legibility* of LLM-generated content as whether the correctness is easily verifiable by humans. They emphasise that legibility during training must sacrifice some correctness.

**Disambiguity Metric.** We propose a *Disambuguity Metric* and *Reliability Score* to achieve a robust decoding scheme that fully incorporates both correctness alignment and ambiguity detection without training. First, we define a Disambiguity Metric:

> **Definition 4.** *(Disambiguity Metric) For a prompt $x$ and a finite set of answer candidate $\mathcal{Y}$, a Disambiguity Metric is a function that projects the prompt $x$ and a candidate $y_i \in \mathcal{Y}$ such that $DA(x, y) : x \times \mathcal{Y} \to [0, 1]$. The output of the function is a measurement of the disambiguation of $y_i \in \mathcal{Y}$ to the prompt $x$. The disambiguity metric, combined with the correctness parameter, can detect false-positive and false-negative classifications in the correctness alignment phase.*

Therefore, for the elements in $\mathcal{Y}$, we assign the correctness parameter from the previous stage as $c(y_i)$, which is the probability distribution of $P_{LM}(y \mid x, \text{correct})$ condition on that the DE is reached. Similarly, we denote the disambiguity metric for $(x, y_i)$ as $DA(x, y_i)$. For any $y_i, y_j \in \mathcal{Y}$, we have $c(y_i) > c(y_j) \cup c(y_i) \leq c(y_j)$ and this also applies for the disambiguity metric $DA(x, y_i), DA(x, y_j)$.

**Disambiguity Maximization.** We introduce the metric $Rel(x, y_i) = \eta \cdot DA(x, y_i) + (1 - \eta)c(y_i)$, reliability, for $\eta \in [0, \overline{\eta}], \overline{\eta} < 1$. $(1 - \overline{\eta})$ is the least proportion of correctness that needs to be considered. This is a convex combination of correctness and disambiguity, thus the maximization problem is defined as: $\max \eta$ s.t. $\min Rel(y_{i,C}) \geq \max Rel(y_{i,I}), \eta < \overline{\eta}$.

> **Intuition 1.** The solution to the maximization problem, denoted as $\eta^*$, is the value that for any $\eta$ such that $\eta^* < \eta < \overline{\eta}$, the preference ordering is different from when $\eta < \eta^*$ which indicates the candidates whose relations with others are altered are the **specious** candidates.

Once we identified the **specious** candidates, the candidates that are the most preferred and the least preferred are the **valid** candidates. The *Reliability* of a disambiguity metric can now be defined for a prompt-candidate set $(x, \mathcal{Y})$.

> **Definition 5.** *(Reliability) A prompt-candidate set $(x, \mathcal{Y})$ couple can be made **more Reliable** by a Disambiguity Metric if such a $\eta^*$ exist for the maximization problem $\max \eta$ s.t. $\min Rel(y_{i,C}) \geq \max Rel(y_{i,I}), \eta < \overline{\eta}$. If such a maximal $\eta$ does not exist, then we say that the prompt-candidate set **cannot be made more Reliable** by Disambiguity Metric.*

> **Theorem 4.** *A prompt-candidate couple can be made **more Reliable** by the disambiguity metric $DA(x, y), y \in \mathcal{Y}$ if and only if (1) $\min c(y_{i,C}) > \max c(y_{i,I})$ and (2) $\overline{\eta} \cdot DA(x, y_{i,I}) + (1 - \overline{\eta})c(y_{i,I}) > \overline{\eta} \cdot DA(x, y_{i,C}) + (1 - \overline{\eta})c(y_{i,C})$ for some $y_{i,C}, y_{i,I}$*

> **Intuition 2.** As for the first condition, the least preferred correct candidate has to be preferred over the most preferred incorrect candidate. Secondly, some incorrect candidates are strictly preferred to some candidates that are initially classified as correct, when disambiguation is maximized. Those two conditions ensure the decoding preference changes under the constraint.

## 3 EXPERIMENTS

We aim to answer the following questions: (1) What design choices enable decoding games to improve language generation performance? (2) To what extent does our BDG improve consistency? (3) To what extent does the BDG improve factual validity and reliability? BDG focuses on improving the consistency and reliability of LLMs. However, consistency and reliability manifest themselves in various forms across different domains and dimensions, including correctness, truthfulness, factuality, valid reasoning, value alignment, among others. In (§3.1), we first assess efficiency and reliability through a multidimensional comparison with another game-theoretic method (Jacob et al., 2024)

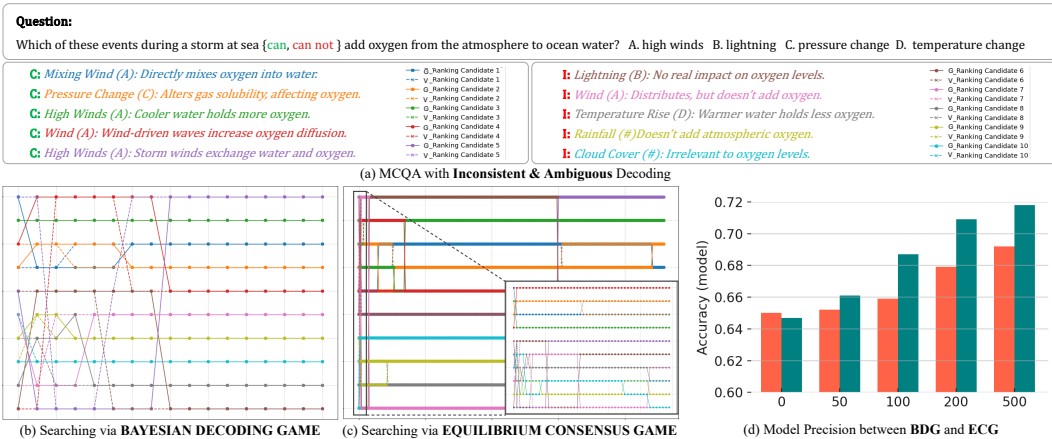

Figure 3: **BDG**'s game design quickly reaches equilibrium and consensus between the generator and discriminator, typically within 100 epochs. In contrast, **ECG** requires significantly more epochs (3000 in this case) and exhibits continuous fluctuations (as shown in the lower right) before achieving consensus. (Zoom in for details.)

and several variants. In (§3.2), we evaluate performance on a diverse set of LLMs used for real-world tasks: MMLU (Hendrycks et al., 2020b), ARC-Easy (E.), -Challenge (C.) (Clark et al., 2018), RACE-High (H.) (Lai et al., 2017). It is important to note that BDG is a game-theoretic decoding strategy and not a deliberation/training-based method like a prover-verifier-game (PVG) (Hendrik Kirchner et al., 2024), or contrastive-objective based generation (Li et al., 2022). Nevertheless, we will demonstrate through benchmarks in reasoning task: GSM8K (Cobbe et al., 2021), medical taks: PubMedQA (Jin et al., 2019), MMLU-Medical (M.), and ethical scenarios, including justice, virtue, deontology and utilitarianism in Ethics (Hendrycks et al., 2020a), that BDG yields significant improvements and demonstrates synergistic potential across various scenarios (§3.3).

**Actions in the Game.** As noted in §2, to adapt BDG to existing methods, a generator in the modeling picks a distribution over a finite set of candidates $\mathcal{Y}$. E.g., in multiple-choice tasks, these are the multiple choice options. In generative tasks, a common approach to generate the finite set of candidates is via sampling with nucleus (Holtzman et al., 2019) and top-k (Fan et al., 2018) from the distribution $P_{\text{LLM}}(y \mid q, \text{correct})$ where $y \in \mathcal{Y}$.

**Baselines and Models.** For fair comparisons, following the setting and scores (Jacob et al., 2024), we use LLaMA models (Touvron et al., 2023) (7B, 13B parameters) with 16-bit inference across all experiments unless otherwise specified. On multiple-choice datasets, we employ: *Generative Ranking (G):* Ranks candidates by $P_{\text{LLM}}(y \mid x, \text{correct})$ following (Brown et al., 2020; Touvron et al., 2023). *Discriminative Ranking (D):* Re-weights query-candidate pairs using $\pi_D^{(1)}(\text{correct} \mid x, y)$ based on (Jacob et al., 2024). *Self-Contrastive Decoding (SCD):* Utilizes $\pi_G^{(1)}$ for reweighting candidates (Jacob et al., 2024; Li et al., 2022). *Equilibrium Consensus Game (ECG):* Reweights pairs with equilibrium discriminator $(x, y)$ by $\pi_D^*(\text{correct} \mid x, y)$ (Jacob et al., 2024). And BDG-based discriminator $(x, y)$ by $\pi_D^*(\text{correct} \mid x, y)$ to reweight query-candidate pairs.

**Prompting.** Unless otherwise specified, the condition for the $P_{LLM}$ corresponds to the standard zero-shot prompt (Jacob et al., 2024; Hendrycks et al., 2020b). Furthermore, we combine chain-of-thought (CoT) (Wei et al., 2022), and few-shots setting (Wei et al., 2022) as orthogonal analysis.

### 3.1 GAME-THEORETIC DESIGN

**Searching & Convergence Behavior.** We first compare searching behaviors of **BDG** with the most closely related method, the **ECG** (Jacob et al., 2024), in the multiple-choice question answering (MCQA) task (Clark et al., 2018). Fig.3 provides a visual case study. BDG demonstrates a swift and consistent convergence in (b).

Table 1: Comparison between ECG and BDG.

| Criteria | ECG: Equilibrium Consensus Game | BDG: Bayesian Decoding Game | Thm. |
|---|---|---|---|
| Strategy | ER-update $x_{i,t+1} = x_{i,t} + \frac{1}{2t}\Sigma_0^t x_{-i,t}$ | last-round belief update $b_{i,t} = a_{-i,t-1}$ | 2 |
| Convergence | No guarantee | Bayes-CCE | 3 |
| Recall | Perfect recall average | Markovian | 3 |
| Complexity | $\mathcal{O}(n^2)$ | $\mathcal{O}(n \log n)$ | N/A |

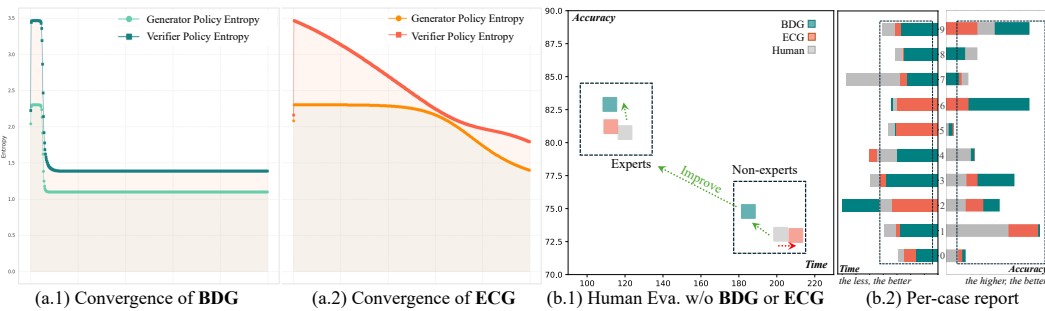

(a.1) Convergence of **BDG** | (a.2) Convergence of **ECG** | (b.1) Human Eva. w/o **BDG** or **ECG** | (b.2) Per-case report

Figure 4: **Entropy dynamics during convergence.** (a.1) Fluctuations in **BDG** indicate exploration of multiple equilibria. (a.2) **ECG** shows persistent entropy fluctuations and continued exploration without reaching stabilization. **BDG improves LLM consistency and reliability for human.** (b.1) Impact of BDG and ECG on time, accuracy for human experts *vs.* non-experts. (b.2) BDG and ECG report on time, accuracy per case for human experts *vs.* non-experts. (Zoom in for details.)

Conversely, the ECG, shown in (c), exhibits prolonged and inconsistent searching behavior. Despite continuous shifts in candidate selections, ECG fails to achieve stable convergence with persistent disagreement between the generator and verifier. (d) and Tab.1 highlights the enhanced and fast convergence properties of the BDG over the ECG.

**Entropy & Equilibrium Convergence.** The staged convergence of BDG with 500 epochs is shown in Fig. 4, which first stabilizes on one solution before shifting to another. This leads to rapid convergence with our no-regret optimization. Compared to the ECG in Fig. 4 (a.2), this phase shows a stabilization in policy entropy, which signifies the model's swift approach to a potential equilibrium. The subsequent shift in policies, resulting in entropy changes, reflects the exploration of multiple PBEs inherent in signaling games. As the BDG navigates these equilibria, entropy fluctuations represent the ongoing search for an optimal balance. The search is further refined by the Decoding Equilibrium, where we design Markovian updating strategies to achieve the best possible alignment between the generator and verifier. In contrast, the ECG tends to keep searching, but lacks the ability to explore, often resulting in a single, less optimal equilibrium.

**Inherent Inconsistency & Reachable Consistency.** We quantified the degree of inconsistency during the decoding stage by analyzing the disagreement percentage between Generative (G) and Discriminative Ranking (D) following (Jacob et al., 2024). In Tab.2, G and D often yield conflicting results, indicating significant

Table 2: Comparison of inconsistency (InC.%) and improvements (Imp.%) between G, ECG, and BDG.

| Domain | Model | InC.% | G | ECG | Imp.% | BDG | Imp.% |
|---|---|---|---|---|---|---|---|
| MMLU | LLaMA-7B | 69.0%↓ | 30.4 | 39.9 | 31.3%↑ | 40.5 | 33.2%↑ |
| | LLaMA-13B | 60.6%↓ | 41.7 | 45.1 | 8.1%↑ | 46.9 | 12.5%↑ |
| ARC-E. | LLaMA-7B | 56.1%↓ | 68.2 | 71.5 | 4.8%↑ | 75.3 | 10.4%↑ |
| | LLaMA-13B | 46.1%↓ | 71.2 | 76.4 | 7.3%↑ | 78.1 | 9.7%↑ |
| ARC-C. | LLaMA-7B | 65.9%↓ | 47.3 | 58.3 | 23.2%↑ | 59.6 | 26.0%↑ |
| | LLaMA-13B | 59.1%↓ | 51.9 | 61.4 | 18.3%↑ | 62.2 | 19.8%↑ |
| RACE-H. | LLaMA-7B | 62.0%↓ | 46.4 | 56.4 | 21.5%↑ | 57.7 | 24.4%↑ |
| | LLaMA-13B | 58.8%↓ | 47.9 | 62.8 | 31.1%↑ | 60.3 | 25.9%↑ |
| **Average** | | 59.7%↓ | 50.6 | 59.0 | 18.2%↑ | 60.1 | 20.2%↑ |

inherent inconsistencies during the decoding stage of generative models. These discrepancies can be effectively mitigated by our approach, specifically during the decoding process, without the need for additional training. Tab. 2 shows that BDG consistently outperforms both G and ECG, particularly in cases with higher disagreement rates. We achieve superior consistency with higher correctness, and reliability with fewer computations in Fig. 3.

**Intrinsic Ambiguity & Provable Reliability.** Shifting our focus from consistency to reliability, we analyze the performance and reliability of game-theoretic decodings in human comprehension tasks. We conducted time-constrained human evaluations *with* and *without* **BDG** and **ECG**. Due to intrinsic ambiguity, human evaluators take longer and are less likely to achieve high accuracy and confidence, where there is a significant disparity in both time and accuracy between experts and non-experts over 200 samples, even the limitations of experts as shown in Fig. 4 (b.1). Game-theoretic approaches enhance the decoding process effectively in a training-free manner, without compromising accuracy. Especially, **BDG** provides a decoding process closely aligned with human intent, improving accuracy consistently for both experts and non-experts, and significantly enhancing the

identification of reliable samples with reduced time in Fig. 4 (b.1). Furthermore, BDG outperformed ECG across various dimensions, as seen in Fig. 4 (b.2). For further details, refer to Appx. H.

## 3.2 Consistency Benchmarking: Across Domains with Smaller Models

With "relatively easy" reasoning and comprehension tasks, we show superior performance compared to baselines and other game-theoretic methods in Tab. 3 due to the efficient alignment of consistency. In a broader comparison, our zero-shot LLaMA-13B (78.1, ARC-E.) outperforms much larger models like the PaLM-540B model (76.6) (Chowdhery et al., 2023).

Table 3: Model consistency across different domains.

| Domain | Model | G | MI | SCD | D | ECG | BDG |
|--------|-------|-----|-----|------|------|------|------|
| MMLU | LLaMA-7B | 30.4 | 33.1 | 30.5 | 40.4 | 39.9 | **40.5** |
| | LLaMA-13B | 41.7 | 41.8 | 41.7 | 41.9 | 45.1 | **46.9** |
| ARC-E. | LLaMA-7B | 68.2 | 68.8 | 69.5 | 52.5 | 71.5 | **75.3** |
| | LLaMA-13B | 71.2 | 71.5 | 73.0 | 65.0 | 76.4 | **78.1** |
| ARC-C. | LLaMA-7B | 47.3 | 47.4 | 56.5 | 42.7 | 58.3 | **59.6** |
| | LLaMA-13B | 51.9 | 52.1 | 59.3 | 48.5 | 61.4 | **62.2** |
| RACE-H. | LLaMA-7B | 46.4 | 46.3 | 53.1 | 46.0 | 56.4 | **57.7** |
| | LLaMA-13B | 47.9 | 48.4 | 58.9 | 55.1 | **62.8** | 60.3 |

With more challenging reasoning and multitask understanding tasks, such as ARC-C, RACE-H, and MMLU, we achieve the best equilibrium decoding with fewer rounds and higher accuracy. Our LLaMA-13B (46.9, MMLU; 57.7, RACE-H.) outperforms zero-shot GPT-3-175B (37.7, MMLU) (Hendrycks et al., 2020b), LLaMA-65B (51.6, RACE-H.) (Touvron et al., 2023), and PaLM-540B (49.1, RACE-H.) (Hendrycks et al., 2020b).

## 3.3 Reliability Gains: Orthogonal Enhancements for Robust Decoding

Datasets in Tab. 4, 5 involve challenging scenarios to test models' reasoning and reliability abilities. We use these benchmarks to study whether we can combine our approach with various orthogonal strategies. As a decoding strategy based on game theory, BDG does not conflict with the computationally intensive game mechanism during training, nor does it conflict with CoT and few-shots based on prompting engineering. We can achieve the improved performance in more challenging scenarios and is an extremely novel decoding research direction with reliable performance. We are also able to achieve wider accuracy and reliability on ethical datasets.

Table 4: The reliability across different domains with CoT.

| Domain | Model | Decoding Methods | | | | Game-theoretic | |
|--------|-------|--------|------|------|------|------|------|
| | | Greedy | MI | SCD | D | ECG | BDG |
| GSM8K | LLaMA-7B | 10.8 | 14.7 | 13.4 | 15.0 | 15.1 | 15.8 |
| | LLaMA-13B | 14.9 | 22.5 | 23.1 | 22.5 | 23.0 | 22.7 |
| TruthfulQA | LLaMA-7B | 33.41 | 34.79 | 34.91 | 34.17 | 34.27 | 35.07 |
| | LLaMA-13B | 33.05 | 36.30 | 34.61 | 39.05 | 38.63 | 40.01 |

Table 5: The reliability with few shots.

| | Domain | Model | BDG | |
|--|--------|-------|-----------|----------|
| | | | zero-shot | few-shot |
| Medical | PubMedQA | LLaMA-7B | 71.45 | 71.89 |
| | | LLaMA-13B | 74.00 | 74.47 |
| | MMLU-M. | LLaMA-7B | 51.35 | 52.90 |
| | | LLaMA-13B | 56.01 | 58.85 |
| Ethics | Justice | LLaMA-13B | 52.27 | 53.15 |
| | Virtue | LLaMA-13B | 33.10 | 33.82 |
| | Deontology | LLaMA-13B | 52.41 | 53.01 |
| | Utilitarianism | LLaMA-13B | 65.35 | 66.75 |

## 4 Discussion

**Game Design over ECG and PVG.** BDG and ECG share the common goal of aligning generative models with human intentions to improve output reliability, yet they differ significantly in their game design, achieving substantial gains with reduced computational overhead. While ECG utilizes moving-average updates to foster consensus, often leading to unstable and fluctuating equilibria, BDG employs a structured Bayesian framework that drives interactions toward an optimal equilibrium with greater stability. In contrast, Prover-Verifier Games (PVGs), which contribute to ChatGPT4-o1, use a RL-based alignment and focus on adversarial training phases featured by RL and competitive dynamics. This requires intensive training and causes potential deviations from cooperative strategies. Appx. E and F explore the distinct phases and transitions between these frameworks, highlighting BDG's scalability and its departure from the training-intensive PVG.

**Robustness and Integrative Potential.** BDG consistently yields improved results, surpassing or matching the performance of benchmark approaches across various domains and scenarios. This robustness is particularly novel, as it demonstrates that BDG is adept at handling diverse scenarios,

even in situations when the initial LLMs are not effective. BDG can also be combined with deliberation methods like self-consistency or prompting methods like CoT. BDG demonstrates computational efficiency by fast equilibrium convergence, and reliable guidance by human evaluation.

**Balancing Correctness and Reliability.** Reliability (Rastogi et al., 2023) tries to give an account of the prover model's failure modes and sense-making, whether the reasoning is correct or not. The resulting decoding can be arbitrarily complex (Nanda et al., 2023). In contrast, correctness allows to verify if a given solution is correct, ignoring how the generator reasoned it to be reliable (consistent with the environment). Consequently, reliability requires model outputs that are coherent and consistent to human understanding (Mökander et al., 2023). We show that it is possible to have both, without sacrificing correctness for reliability (Hendrik Kirchner et al., 2024), and especially in high-stakes settings reliability is as important as correctness (Casper et al., 2024).

**Limitation.** One potential limitation arises from the explicit specification of correctness consistency branches during the game process, as this alignment is primarily intended to match human intent with model outputs, similar to game-based approaches (Jacob et al., 2024; Hendrik Kirchner et al., 2024). While this structure ensures consistency, it may introduce a subtle bias by prioritizing accuracy. However, in our second stage, this bias can be mitigated through further calibration and other orthogonal strategies, offering more refined guidance to human evaluators. Adding multi-metrics and multiple agents to achieve game-based deliberation will be a future development.

## 5 RELATED WORK

**Multi-Agent Debate Frameworks.** Previous work has explored mechanisms where multiple language model instances "debate" to refine and converge to a final answer (Du et al., 2023; Chen et al., 2023; Khan et al., 2024). It is possible to see our method as a major variant of this multi-agent debate in which the interaction occurs within a game-theoretic framework, rather than directly within the language models' outputs. This structured signaling game enables *BDG* to enhance the correctness and reliability of outputs without relying on human feedback, by dynamically optimizing the generation and verification processes. Additionally, this approach can resolve ambiguity, confusion, and low accuracy caused by inconsistencies, but not by poor reasoning.

**Decoding Strategies.** Top-k sampling (Fan et al., 2018), nucleus sampling (Holtzman et al., 2019), and typical sampling (Meister et al., 2023) focus on generating high-confidence text but do not address the correctness of the outputs. Candidates were generated using these methods. Equilibrium-ranking (Jacob et al., 2024) applies an average-moving strategy to the initial distribution. In contrast, *BDG* integrates a multistage signaling game that inherently balances correctness and consistency during the generation process. BDG can be seamlessly combined with these strategies to enhance the reliability and reliability of generated text.

**Ranking Techniques.** Rranking is a widely used approach to select the correct output from a set of candidates generated by language models. (Thoppilan et al., 2022) use additional human annotations to train a ranking model for response filtering. (Hendrik Kirchner et al., 2024) trains different provers and verifiers for increasing output legibility. Although our work also utilizes existing language models as discriminators, *BDG* eliminates the need for additional training and does not impose specific assumptions on the structure of either the generator or discriminator.

## 6 CONCLUSION

The Bayesian Decoding Game (BDG) is a novel game-theoretic framework that enhances both the consistency and reliability of LLMs. By framing the decoding process as a multistage signaling game between a generator and verifier, BDG efficiently aligns model outputs with human intent while mitigating the trade-off between correctness and reliability. Our approach achieves superior performance across benchmarks, often surpassing larger models, and demonstrates its adaptability when combined with existing techniques like chain-of-thought prompting. BDG ensures reliable and robust LLM outputs, offering a scalable, training-free solution to the challenges of ambiguity and inconsistency in generative models.

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

# Appendix Contents

## A    REPRODUCIBILITY STATEMENT

We conducted our evaluations using widely recognized benchmarks such as ARC-Easy, ARC-Challenge, MMLU, and RACE. The experiments were performed using the open-source LLaMA 7B and 13B models. Key aspects of the game, including update policies and initial strategies, are thoroughly detailed in both the main text and appendix to facilitate accurate replication of the results. All experiments were conducted on NVIDIA A6000 and A100 GPUs, with runtimes ranging from 0.5 to 6 hours depending on the model size, task, and experimental settings. Further details on the game-theoretic mechanisms and specific design choices can be found in the methods section and the appendix.

## B    POTENTIAL ETHICS RISKS AND SOCIETAL IMPACT

Bayesian Decoding Game (BDG) is a novel game-theoretic framework that significantly enhances both the consistency and reliability of large language model outputs. By framing the decoding process as a multistage signaling game between a generator and verifier, BDG efficiently aligns model outputs with human intent while mitigating the trade-off between correctness and reliability. BDG ensures reliable and robust LLM outputs, offering a scalable, training-free solution to the challenges of ambiguity and inconsistency in generative models.

With the improvement of generation quality, one can imagine more potent disinformation (e.g., automatic generation of fake news) that may be hard to distinguish from human-authored content. It might be worthwhile to augment current decoding techniques so that the generated outputs will also be watermarked without compromising their quality.

## C    GAME-THEORETIC FORMULATION SUPPLEMENTARY

A generative language model (LM) maps input $x$ to output $y$ according to some distribution $P_{\text{LM}}(y \mid x)$. Here, we do not impose restrictions on the form of input or output, as illustrated in Fig.1. Instead, we address a multi-faceted problem involving a question $x$ and a set of answer candidates $\mathcal{Y}$, generated by pre-trained language models on specific tasks. In the first stage, using this candidate set, we leverage generative LMs in two distinct ways:

*Generatively*, by supplying as input

1. a prompt $x$,

2. the set of candidates $\mathcal{Y}$, and

3. a natural language prompt indicating that a correct or incorrect answer is desired. The LM may be thought of as modeling a distribution $P_{\text{LM}}(y \mid x, \text{incorrect})$, where the token incorrect denotes the fact that the model was prompted to generate an incorrect answer.

*Verifiably*, by supplying as input

1. the same $x$ and

2. a possible candidate answer $y \in \mathcal{Y}$, together with

3. a prompt indicating that a correctness assessment $v \in \{\text{correct, incorrect}\}$ is sought. In this case, the language model acts as a models a distribution $P_{\text{LM}}(v \mid x, y)$ where $v \in \{\text{correct, incorrect}\}$.

The essence of a signaling game (Gibbons et al., 1992) is that one player (the generator) takes an action, the signal, to convey information to another player (the verifier); in the simplest setup, the final payoff depends on whether the verifier correctly judges the generator's type based on the generator's signal. Based on this intuition from game theory, (Jacob et al., 2024) design a Equilibrium Consensus Game (ECG), without a formal definition of the game. Thus, we firstly provide a comprehensive game-theoretic formulation for generative model decoding, and propose improvements to address limitations.

Formally, the signaling game's components can be defined as follows:

1. *Players*: Generator and Verifier;

2. *Choice sets*: Generator's choice set is $y \in \mathcal{C}_G = \mathcal{Y}$, with prompt $p$ randomly drawn from {Correct, Incorrect}, and the Verifier's choice set is $v \in \mathcal{C}_V = $ {Correct, Incorrect}, based on the generator's choice $y \in \mathcal{Y}$;

3. *Payoff Function*: $u_G = u_V = \mathbb{1}_{p=v}(p, v)$, where $\mathbb{1}$ equals 1 if the correctness prompt $x$ matches the verification result, and 0 otherwise.

We are now ready to state the fundamental concept of this signaling game, a Perfect Bayesian Nash Equilibrium (PBNE) (Cho & Kreps, 1987). We use the short form Perfect Bayesian Equilibrium (PBE) with the auxiliary definitions Defi. 6. and 7. for PBE Definition.

**Definition** 1. (Perfect Bayesian Equilibrium (Fudenberg, 1991)) A Perfect Bayesian Nash Equilibrium (PBE) is a pair $(s, b)$ of strategy profile and a set of beliefs such that

1. $s$ is **sequentially rational** given beliefs b, and

2. $b$ is **consistent** with $s$.

> **Example 1.** For generative model decoding, the generator's belief is given by its perceived probability distribution, $\mathbb{P}(\{\textbf{correct}, \textbf{incorrect}\}) = (p_i, 1 - p_i)$, for each $y_i \in \mathcal{Y}$ of the verifier's judgment, and with its belief and type, the generator chooses a mixed action that maximizes its utility, i.e., if the generator's type is **correct**, then its optimal mixed strategy would be allocating positive possibility only on $y_i$ such that $p_i > 1 - p_i$ and zero possibility to other $y_i$.

**Definition 6.** *(Sequential Rationality)*
*A player is said to be sequentially rational iff, at each information set he is to move, he maximizes his expected utility given his beliefs at the information set (and given that he is at the information set) - even if this information set is precluded by his own strategy.*

**Definition 7.** *(Consistency on Path)*
*Given any (possibly mixed) strategy profile $s$, an information set is said to be on the path of play if and only if the information set is reached with positive probability according to $s$. Given any strategy profile $s$ and any information set $I$ on the path of play of $s$, the beliefs of a player at $I$ are said to be consistent with $s$ if and only if his beliefs are derived using the Bayes rule and s.*

## D    PROOFS OF THEOREMS

### D.1    PROOF OF THEOREM 1

**Theorem** 1. More than one (mixed) strategy Perfect Bayesian Equilibrium exists for this game.

***Proof of Theorem 1.:***

Suppose that the candidate set has 2 options (can be extended to any cardinality $|\mathcal{Y}|$), $y_1, y_2$, one equilibrium can be described as: If the environment sends correct/incorrect, the generator generates the probability distribution $(1, 0)/(0, 1)$ for $(y_1, y_2)$ given his belief that verifier probabilistic judgment, {correct, incorrect}, for $y_1, y_2$ is $(1, 0), (0, 1)$.

For the verifier, he believes that if the environment chooses correct/incorrect, then he believes that of generator's probabilistic generation for $(y_1, y_2)$ are $(1, 0), (0, 1)$, therefore the verifier's best response is given by (correct, incorrect) = $(1, 0)$ if sees $y_1$, (correct, incorrect) = $(0, 1)$ if sees $y_2$. The (action and belief) for the generator and verifier above constitute one PBE for our game. For another equilibrium, we can revert every 0s and 1s in the above strategy profile, for all the actions and the beliefs.

### D.2    PROOF OF THEOREM 2

**Theorem** 2. There are $n!$ number of Equilibria for the Decoding Game.

***Proof of Theorem 2.:***

Since there are $n!$ permutations of elements in $\mathcal{Y}$, there are $n!$ DEs for the decoding game.

To accommodate the refined utility we proposed, we introduce another definition such that if

$$\|a_{\mathrm{G}}(y \mid x, \text{correct}, b_S) - a_{NV}(\text{correct} \mid x, y, b_V)\| < \sigma$$

for every $y$, where $a_{NV}$ is the normalized probability distribution of the verifier's action, we call it a $\sigma$-DE.

### D.3 PROOF OF THEOREM 3

**Theorem** 3. The Markovian update schedule for our Decoding Game will converge to an equilibrium.

***Proof of Theorem 3.:***

We will show that the Markovian update schedule is in fact no-regret(thus guarantees CCE-convergence) for correct generator, and when generator receives incorrect signal, she will automatically perform the reversed action; then, if the Markovian update schedule converges to CCE for the incorrect signal, it automatically satisfies that the Markovian schedule will converge to a Bayes-CCE of our Decoding Game.

**Definition 8.** *A randomized strategy profile* $\mathbf{s} \in \Delta(\Sigma)$ *is a coarse-correlated Bayesian equilibrium if for every* $a_i' \in A_i$ *and for every* $v_i \in \mathcal{V}_i$ *:*

$$\mathbb{E}_{\mathbf{s}}\mathbb{E}_{\mathbf{v}}\left[U_i\left(\mathbf{s}(\mathbf{v}); v_i\right) \mid \mathbf{v}_i = v_i\right] \geq \mathbb{E}_{\mathbf{s}}\mathbb{E}_{\mathbf{v}}\left[U_i\left(a_i', \mathbf{s}_{-i}\left(\mathbf{v}_{-i}\right); v_i\right) \mid \mathbf{v}_i = v_i\right]$$

We will first prove that the Markovian update schedule is asymptotically no-regret. For the generator, suppose that at time $t$, the chosen action is $a_G^{(t)}(y \mid x, \text{correct})$, and the optimal hindsight action that maximize $U$ is given by $a_{NV}^{(t)}(\text{correct} \mid x, y) \forall y$, which is the normalized verifer's action on each candidate $y \in \mathcal{Y}$, and our update schedule

$$a_{\mathrm{G}}^{(t+1)}(y \mid x, v) \propto \exp\left\{\frac{\frac{1}{2}b_{\mathrm{G}}^{(t+1)}(y \mid x, v) + \lambda_{\mathrm{G}} \log a_{\mathrm{G}}^{(t)}(y \mid x, v, b_{\mathrm{G}}^{(t)})}{1/(\eta_{\mathrm{G}}t) + \lambda_{\mathrm{G}}}\right\}$$

such that

$$b_{\mathrm{G}}^{(t+1)}(y \mid x, v) = a_{\mathrm{V}}^{(t)}(v \mid x, y)$$

therefore, the regret at time $t$ is given by

$$\|a_G^{(t)}(\mathbf{y} \mid x, \text{correct}) - a_{NV}^{(t)}(\text{correct} \mid x, \mathbf{y})\|$$

and in time $t+1$, we have that the generator is **at least** $\frac{1}{2}\eta_G$ closer to the verifier's action, and the verifier is also **at least** $\frac{1}{2}\eta_V$ closer to the generator's action. Thus we have that

$$\|a_G^{(t+1)}(\mathbf{y} \mid x, \text{correct}) - a_{NV}^{(t+1)}(\text{correct} \mid x, \mathbf{y})\| \leq$$
$$\left(1 - \frac{1}{2}(\eta_G + \eta_V)\right)\|a_G^{(t)}(\mathbf{y} \mid x, \text{correct}) - a_{NV}^{(t)}(\text{correct} \mid x, \mathbf{y})\|$$

and by construction we have $1 - \frac{1}{2}(\eta_G + \eta_V) < 1$, then asymptotically we can obtain that

$$\lim_{t \to \infty} \|a_G^{(t)}(\mathbf{y} \mid x, \text{correct}) - a_{NV}^{(t)}(\text{correct} \mid x, \mathbf{y})\| \to 0$$

thus the cumulative average regret also goes to 0 asymptotically. Therefore, the Markovian update schedule is no-regret and thus will converge to a Bayes-CCE of this game.

## E FROM **TRAINING-FREE** BAYESIAN DECODING GAME (BDG) TO **RL-BASED** PROVER-VERIFIER GAME(PVG)

Prover-Verifier Game (PVG) (Hendrik Kirchner et al., 2024), structured as zero-sum games, encounter substantial challenges that undermine their efficacy in ensuring reliable outputs. The adversarial nature of zero-sum games inherently prioritizes winning over mutual consistency, which

leads to strategic behavior focused on exploiting the opposing agent rather than achieving genuine correctness e.g., model collapse. This often results in provers generating outputs that are optimized to mislead the verifier rather than to align with factual truth, thus producing equilibria that favor strategic manipulation over accurate assessment. Such dynamics complicate the training process, requiring extensive tuning and computational resources without guaranteeing robust, interpretable results. Furthermore, the reliance on reinforcement learning in these systems falls short of effectively replacing human feedback, as the trained verifier cannot fully replicate the nuanced judgment required to evaluate complex or ambiguous output. These limitations fall into the misalignment between training objectives and practical needs, where models become adept at adversarial optimization but lack the reliability and consistency necessary for real-world application. However, Bayesian Decoding Game (BDG) can bring the advantages of the game and bridge the purposes with proper implementations, which bypass the extensive training and adversarial pitfalls by directly modeling output verification through probabilistic reasoning, enhancing the interpretability and reliability of the generated content without the dependencies on zero-sum competition.

Here, we consider the connections between the PVG and BDG, and before the comparison, we give a brief introduction of PVG.

We consider a scenario of problems with ground-truth answers $(x, y) \in \mathcal{X} \times \mathcal{Y}$, where $x$ denotes a problem statement and $y = y(x)$ denotes the mapping from the prompt $x$ to ground-truth answer or set of answers. PVG has set the prover's type to be drawn from a known distribution, which can be either honest or sneaky, to produce an answer $z$ condition on the prompt $x$ and type; such that the honest prover is told to produce correct answers and make sure the verifier recognize the correctness, whereas the sneaky prover is told to trick the verifier by providing wrong answers that are not easy to be detected; then, after receiving solution, the verifier verifies the answer based solely on the answer and prompt, and the output $v(x, z) \in \{0, 1\}$ estimates the correctness of the solution's answer. The output of the verifier is either 1(the statement is correct) or 0(the statement is incorrect).

More precisely, The game follows the procedure below:

1. The type of prover $t$ is randomly drawn from a known distribution $P(t)$, such that $t \in \{h, s\}$, where $h, s$ dictates honest and sneaky prover's type.

2. Given a problem statement, the prover generates proofs by maximizing the payoff function(utility) of each round myopically, such that

$$z = \operatorname{argmax} r(v(x, z), \mathbb{1}_{t=h} \mathbb{1}_{\text{correct}}(x, z) + \mathbb{1}_{t=s}(1 - \mathbb{1}_{\text{correct}}(x, z)))$$

the function $r(v, q)$ satisfies that $r(1, 1) > r(0, 1) > r(1, 0) > r(0, 01)$, which corresponds to the fact that both prover prioritizing generating proofs that match their types (correct answer for honest and incorrect answer for sneaky), $\forall v \quad r(v, 1) > r(v, 0)$ and then both want their answers to be accepted as correct by the verifer $\forall q \quad r(1, q) > r(0, q)$.

3. the verifier provides verification $v \in \{0, 1\}$ based solely on the proof $x$ given that the utility is 1 if $v(x, z) = \mathbb{1}(x, z)$ and 0 otherwise
   Theoretically, this is a simple Stackleberg game such that an equilibrium exists and before diving into the equilibrium strategy profile, an important definition from (Hendrik Kirchner et al., 2024) needs to be mentioned.

**Definition 9.** *(Checkability Hendrik Kirchner et al. (2024)) Prover $\pi$ from model family $\mathcal{H}$ is verifiable to verifier $v$ on the data distribution $P_{XY}$ if and only if the prover always generates correct and convincing solutions, and no sneaky prover from the same model family $\mathcal{H}$ can fool the verifier into accepting incorrect solutions. In mathematical notations, the two properties are given by:*
*Completeness: $\forall x \in \mathcal{X}, \mathbb{1}_{correct}(x, \pi(x)) = v(x, \pi(x)) = 1$.*
*Soundness: $\forall \pi' \in \mathcal{H}, \forall x \in \mathcal{X}, \mathbb{1}_{correct}(x, \pi'(x)) = 0 \Longrightarrow v(x, \pi'(x)) = 0$.*

With the definition in hand, the equilibrium strategy profile is given by *(the honest prover always provides checkable and correct proof, the sneaky prover always provides noncheckable and incorrect*

*proof, the verifier can always verify the correctness of the given proof).* For the neural networks to approximate the equilibrium strategies, (Hendrik Kirchner et al., 2024) utilized a reinforcement learning-based algorithm to train the prover and the verifier.

RL-based PVG Hendrik Kirchner et al. (2024) can fit in part into the framework of our training-free BDG framework. As for the game-theoretic setting, PVG is a zero sum verifier-lead Stackleberg game, the strategy update schedule must be modified to fit the utility defined in Hendrik Kirchner et al. (2024). Moreover, the verifier's strategy update cannot be achieved training-free as her utility only depends on the ground truth right/wrong of the candidate and thus needs to be trained; but on the prover side, both honest and sneaky prover can update strategies pain-free from the verifier's trained actions.

Firstly, we define the strategy for verifier and prover in the same way as in BDG, such that given the environment signal, the prover generates a probability distribution for a set of answers, and the verifier always generates a probability distribution of {correct, incorrect} for each of the answers. Also, we make the same assumption that each player can observe the opponent's full action profile rather than the realized action; then, we are ready to highlight the difference in schedule update under the Markovian schedule, the condition where $v = $ correct, we will abbreviate that as correct $= \mathbf{C}$, stays the same, such that because they want to align their actions with the verifier

$$b_{\mathrm{P}}^{(t+1)}(y \mid x, \mathbf{H}) = a_{\mathrm{V}}^{(t)}(\mathbf{C} \mid x, y)$$

$$a_{\mathrm{P}}^{(t+1)}(y \mid x, \mathbf{H}) \propto \exp\left\{\frac{\frac{1}{2}b_{\mathrm{P}}^{(t+1)}(y \mid x, \mathbf{C}) + \lambda_{\mathrm{P}} \log a_{\mathrm{P}}^{(t)}(y \mid x, \mathbf{C}, b_{P}^{(t)})}{1/(\eta_{\mathrm{P}}t) + \lambda_{\mathrm{P}}}\right\}$$

However, for the sneaky prover, her utility is maximized when the verifier mistakens the correctness of the problem. Therefore, the optimal update schedule for sneaky prover is given updating toward a normal distribution over the preference generated by the probability distribution of verifier's action. The reason for this update is because near the correct/incorrectness boundary is where the verifier tends to make mistakes, such that

$$a_{\mathrm{P}}^{(t+1)}(y \mid x, \mathbf{S}) \propto \exp\left\{\frac{\frac{1}{2}\mathcal{N}(y \mid a_V) + \lambda_{\mathrm{P}} \log a_{\mathrm{P}}^{(t)}(y \mid x, \mathbf{I}, b_{P}^{(t)})}{1/(\eta_{\mathrm{P}}t) + \lambda_{\mathrm{P}}}\right\}$$

For example, if there are 10 answer candidates, the verifier's preference from her action is given by $y_3 \succ y_7 \succ y_6 \succ y_5 \succ y_10 \succ y_2 \succ y_9 \succ y_4 \succ y_1 \succ y_8$, then $\mathcal{N}(y \mid a_V)$ is given by

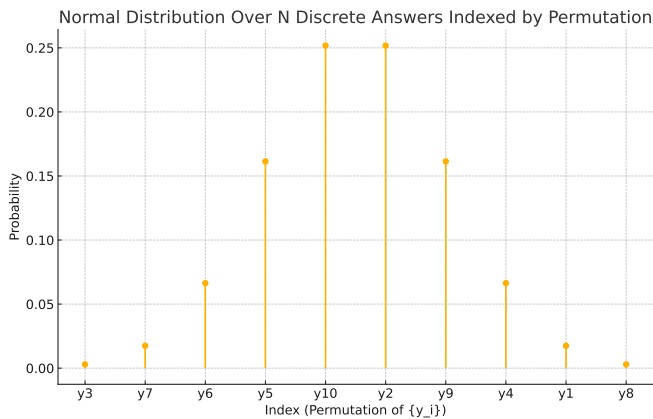

## F FROM MEMORYLESS BAYESIAN DECODING GAME (BDG) TO MOVING-AVERAGE EQUILIBRIUM CONSENSUS GAME (ECG)

The moving average update schedule proposed by Jacob et al. (2024) requires both the generator and the verifier to keep track of the average action of the opponent in addition to the action in the last round, while our Markovian framework allows the players to be memoryless. To better compare

ECG with our update schedule, we provide a general, unifying framework called the History window schedule, where the player's belief is given by the average of past history actions for the period $n$, and at the same time, this schedule retains a large part the initial policy for each round with a stiffness parameter $\lambda_i, i \in \{G, V\}$. The belief is given by

$$b_{\text{G}}^{(t+1)}(y \mid x, v) = \frac{1}{n} \sum_{\tau=t-n+1}^{t} a_{\text{V}}^{(\tau)}(v \mid x, y)$$

$$b_{\text{V}}^{(t+1)}(v \mid x, y) = \frac{1}{n} \sum_{\tau=t-n+1}^{t} a_{\text{G}}^{(\tau)}(y \mid x, v)$$

(2)

Thus the strategy update is given by

$$a_{\text{G}}^{(t+1)}(y \mid x, v) \propto \exp\left\{ \frac{\frac{1}{2} b_{\text{G}}^{(t+1)}(y \mid x, v) + \lambda_{\text{G}} \log a_{\text{G}}^{(1)}(y \mid x, v)}{1/(\eta_{\text{G}} t) + \lambda_{\text{G}}} \right\}$$

$$a_{\text{V}}^{(t+1)}(v \mid x, y) \propto \exp\left\{ \frac{\frac{1}{2} b_{\text{V}}^{(t+1)}(y \mid x, v) + \lambda_{\text{V}} \log a_{\text{V}}^{(1)}(v \mid x, y)}{1/(\eta_{\text{V}} t) + \lambda_{\text{V}}} \right\}$$

As it can be noted in 2, if we take $n = t$, the update schedule coincides with ECG which requires the memory of the moving-average of full history, rather if we take $n = 1$, the update schedule becomes fully memoryless and requires no memory of any past events other than the last period's opponent action.

## G  Experiment Details

**Baselines and Models.** For the fair comparision following(Jacob et al., 2024), we use the same public 7B and 13B parameter models from the LLaMA family(Touvron et al., 2023) and perform 16-bit inference for all our experiments. Since we have a multi-round optimization game and in order to distinguish consensus/ zero-sum games, we define ours as a verifier rather than a discriminator. Across the experiments, all the approaches and orthogonal techniques involved:

- **Generative Ranking (G):** The baseline(Brown et al., 2020; Touvron et al., 2023) ranks every candidate $y$ by $P_{\text{LLM}}(y \mid x, \text{correct})$ and picks the top candidate. This is the standard approach used in past work. Due to implementational differences and non-public resources, we report the existing scores in (Jacob et al., 2024).

- **Discriminative Ranking (D):** Following(Jacob et al., 2024), this approach reweights every query-candidate pair $(x, y)$ by $\pi_D^{(1)}(\text{correct} \mid x, y)$. Typically, this would surpass the performance of ordinary individuals, who might neglect to notice the ambiguity errors. And outstrip the generators that might trust the unreliable decodings.

- **Mutual Information Ranking (MI):** The mutual-information based baseline reweights every candidate $y$ by $P_{\text{LM}}(y \mid x, \text{correct}) \cdot P_{\text{LM}}(\text{correct} \mid x, y)$ (Li & Jurafsky, 2016).

- **Self-Contrastive Decoding (SCD):** The contrastive-based method (Jacob et al., 2024; Li et al., 2022) utilizes the contrastive-based generator $\pi_G^{(1)}$ to reweight every candidate $y$ by $\pi_G^{(1)}(\text{correct} \mid x, y)$. This method achieves a contrasting effect by comparing negative samples instead of employing a verifier (in BDG)/ discriminator (in ECG).

- **Equilibrium Consensus Discriminator (ECG):** This approach is based on discriminator $\pi_D^*$ (Jacob et al., 2024). It reweighs every query-candidate pair $(x, y)$ by $\pi_D^*(\text{correct} \mid x, y)$. This method, involving comprehensive policies and updates, serves as our main benchmark.

- **Bayesian Decoding Game (BDG):** This approach utilizes our Bayesian Decoding Game-based discriminator $\pi_D^*$. This approach reweighs every query-candidate pair $(x, y)$ by $\pi_D^*(\text{correct} \mid x, y)$.

**Orthogonal Techniques.** Furthermore, BDG can combine chain-of-thought (CoT) (Wei et al., 2022) and few-shots setting (Wei et al., 2022) as orthogonal extra gains.

- **Chain-of-Thought (CoT):** CoT (Wei et al., 2022) prompting enables language models to generate intermediate reasoning steps, improving performance on complex tasks. By providing exemplars of reasoning chains, the model is guided to produce more coherent and accurate responses.

- **Few-Shot:** Few-shot setting (Wei et al., 2022) involves providing the model with a small number of example input-output pairs within the prompt. This technique helps the model adapt to the task at hand without additional fine-tuning, improving its ability to generalize from limited data.

**Hyperparameters.** We set $\eta_D$, $\lambda_D$ and $\eta_G$, $\lambda_G$ with 0.1 compared to ECG. Experiments are run 5000 times with early stopping based on equilibrium convergence. BDG can usually converge by 500 iterations or less. The hyperparameters can be larger according to the tasks and initial model ability.

**Extra Metrics.** Following (Li et al., 2022), we have

- *Diversity.* This metric aggregates n-gram repetition rates:

$$\text{DIV} = \prod_{n=2}^{4} \frac{\text{unique n-grams}(x_{\text{cont}})}{\text{total n-grams}(x_{\text{cont}})}.$$

  Models that score low for diversity are prone to repetition, while models that score high for diversity are lexically diverse.

- *MAUVE.* MAUVE (Pillutla et al., 2021) measures the similarity between generated text and gold reference text.

- *Coherence.* (Su et al., 2022) approximates coherence by cosine similarity between the sentence embeddings of prompt $x_{\text{pre}}$ and generated continuation $x_{\text{cont}}$:

$$\text{COH}(x_{\text{cont}}, x_{\text{pre}}) = \frac{\text{EMB}(x_{\text{pre}}) \cdot \text{EMB}(x_{\text{cont}})}{\|\text{EMB}(x_{\text{pre}})\| \cdot \|\text{EMB}(x_{\text{cont}})\|},$$

  where $\text{EMB}(x)$ represents the pre-trained SimCSE embedding (Gao et al., 2021).

- *Human Evaluation.* To further evaluate the quality of the generated text, we consider two critical aspects: *correctness* and confidence in *reliability*. More details can be found in the next section.

## H  HUMAN EVALUATION

**Setting.** In this experiment, participants were tasked with evaluating the correctness of ten answers to a high-school level multiple-choice mathematics problem generated by a Large Language Model (LLM). Participants were instructed to classify each answer as correct, incorrect, or ambiguous. The experiment was conducted in two stages:

In the first stage, participants were given two minutes to classify as many answers as possible, and their results were recorded. In the second stage, participants were allowed to allocate their time freely to complete the remaining classifications, and they were asked to record the time upon completion of their classifications. Below is the questionnaire we utilized for the experiment.

Each participant was randomly assigned three distinct problems, and the corresponding solutions were classified under three conditions: without any hints, with a BDG hint, and with a ECG hint. The hints provided were rankings of the answers generated by the respective models (BDG and ECG). The assignment of different problems across the three conditions was designed to prevent memorization and to control for potential confounding effects related to the content of the specific problem. Problems were drawn from a pool of questions with similar difficulty levels, allowing for consistent observation of treatment effects across varying problem sets.

**Samples.** To better illustrate the experiment setting details, we provided the questionnaire interface, instruction and 2 cases sets below.

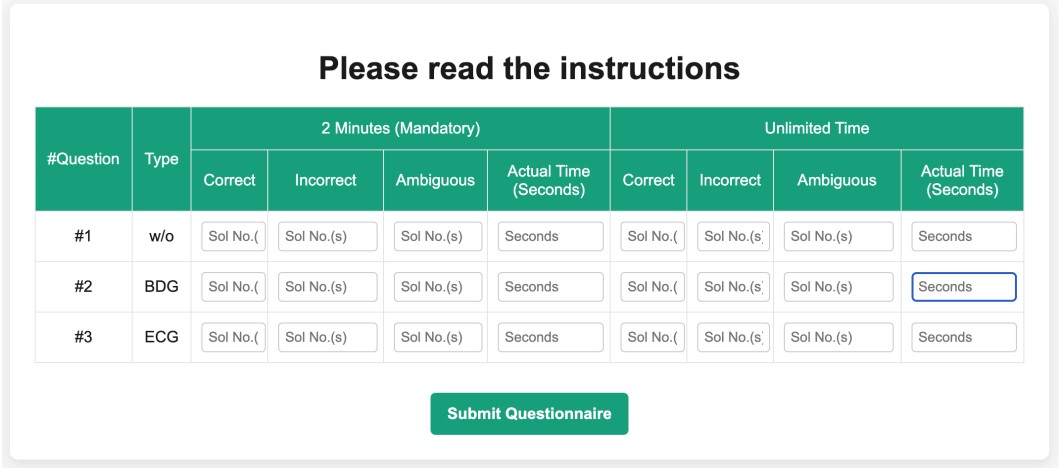

| | |
|---|---|
| **Instructions** | **For task 1**, you will judge the **correctness** of each solution under a mandatory 2-minute limit, and for the ones that you are unsure, label them as ambiguous and record the result on the answer sheet provided. If you did not finish in 2 minutes, then record your incomplete result at the 2-minute cutoff and continue your task until finished, also remember to record your completion time.
**For task 2, 3**, we will provide a reference (ranking of the solutions to assist and accelerate your correctness judgement, however, always stick to your own judgement if there exists a divergence. All other procedures are the same as task 1. |

**Question #**

*Random selected*  On a Cartesian coordinate plane, points $(1,2)$ and $(7,4)$ are opposite vertices of a square. What is the area of the square?

Choices    A. 20        B. 34        C. 40        D. $\sqrt{40}$

Task 1    *Candidates*

0.    "To find the area of the square, we multiply the length of the diagonal, which is $\sqrt{40}$, by itself. This gives us 40, so the answer is C. 40."

1.    "The distance between the points $(1,2)$ and $(7,4)$ is $\sqrt{40}$. This is the diagonal of the square, so the area could either be 20 or 40. The answer might be A. 20."

2.    "The diagonal of the square is $\sqrt{40}$, and the area must be equal to this value. The answer is D. $\sqrt{40}$."

3.    "First, calculate the diagonal between the points $(1,2)$ and $(7,4)$, which gives $\sqrt{40}$. Since this is the diagonal of the square, the area is $\frac{(\sqrt{40})^2}{2} = 20$. Therefore, the answer is A. 20."

4.    "The area of the square is found by calculating the distance between the points. The diagonal is $\sqrt{40}$, so the area is likely $\sqrt{40}$. The answer is D. $\sqrt{40}$."

5.    "First, find the distance between $(1,2)$ and $(7,4)$ using the distance formula: $d = \sqrt{(7-1)^2 + (4-2)^2} = \sqrt{40}$. This distance is the diagonal of the square. The area of the square is then $\frac{(\sqrt{40})^2}{2} = 20$. Therefore, the correct answer is A. 20."

6.    "Using the distance formula, the diagonal of the square is $\sqrt{40}$. Therefore, the area of the square is 20. The answer is A. 20."

7.    "First, calculate the distance between the points $(1,2)$ and $(7,4)$, which gives the diagonal of the square. The distance formula gives $d = \sqrt{(7-1)^2 + (4-2)^2} = \sqrt{40}$. Since the diagonal of the square is $\sqrt{40}$, the area is $(\sqrt{40})^2/2 = 20$. Therefore, the answer is A. 20."

8.    "Using the distance formula, the diagonal of the square is found to be $\sqrt{40}$. The area of the square is 20, so the answer is A. 20."

9.    "Calculate the diagonal of the square using the distance formula: $d = \sqrt{40}$. Since the diagonal is $\sqrt{40}$, the area of the square is simply 40. Therefore, the answer is C. 40."

Task 2    **BDG Guidance** [Will be hidden during evaluation]

0.

*Consistent*    "First, find the distance between $(1,2)$ and $(7,4)$ using the distance formula: $d = \sqrt{(7-1)^2 + (4-2)^2} = \sqrt{40}$. This distance is the diagonal of the square. The area of the square is then $\frac{(\sqrt{40})^2}{2} = 20$. Therefore, the correct answer is A. 20."

1. "First, calculate the diagonal between the points $(1, 2)$ and $(7, 4)$, which gives $\sqrt{40}$. Since this is the diagonal of the square, the area is $\frac{(\sqrt{40})^2}{2} = 20$. Therefore, the answer is A. 20."

2. "Using the distance formula, the diagonal of the square is $\sqrt{40}$. Therefore, the area of the square is 20. The answer is A. 20."

3. "Using the distance formula, the diagonal of the square is found to be $\sqrt{40}$. The area of the square is 20, so the answer is A. 20."

4. "The distance between the points $(1, 2)$ and $(7, 4)$ is $\sqrt{40}$. This is the diagonal of the square, so the area could either be 20 or 40. The answer might be A. 20."

5. "First, calculate the distance between the points $(1, 2)$ and $(7, 4)$, which gives the diagonal of the square. The distance formula gives $d = \sqrt{(7-1)^2 + (4-2)^2} = \sqrt{40}$. Since the diagonal of the square is $\sqrt{40}$, the area is $(\sqrt{40})^2/2 = 20$. Therefore, the answer is A. 20."

6. "The diagonal of the square is $\sqrt{40}$, and the area must be equal to this value. The answer is D. $\sqrt{40}$."

7. "The area of the square is found by calculating the distance between the points. The diagonal is $\sqrt{40}$, so the area is likely $\sqrt{40}$. The answer is D. $\sqrt{40}$."

8. "Calculate the diagonal of the square using the distance formula: $d = \sqrt{40}$. Since the diagonal is $\sqrt{40}$, the area of the square is simply 40. Therefore, the answer is C. 40."

9.

*Inconsistent* "To find the area of the square, we multiply the length of the diagonal, which is $\sqrt{40}$, by itself. This gives us 40, so the answer is C. 40."

---

**Task 3**  **ECG Guidance** [Will be hidden during evaluation]

0.

*Consistent* "First, find the distance between $(1, 2)$ and $(7, 4)$ using the distance formula: $d = \sqrt{(7-1)^2 + (4-2)^2} = \sqrt{40}$. This distance is the diagonal of the square. The area of the square is then $\frac{(\sqrt{40})^2}{2} = 20$. Therefore, the correct answer is A. 20."

1. "Using the distance formula, the diagonal of the square is $\sqrt{40}$. Therefore, the area of the square is 20. The answer is A. 20."

2. "First, calculate the diagonal between the points $(1, 2)$ and $(7, 4)$, which gives $\sqrt{40}$. Since this is the diagonal of the square, the area is $\frac{(\sqrt{40})^2}{2} = 20$. Therefore, the answer is A. 20."

3. "The distance between the points $(1, 2)$ and $(7, 4)$ is $\sqrt{40}$. This is the diagonal of the square, so the area could either be 20 or 40. The answer might be A. 20."

4. "Using the distance formula, the diagonal of the square is found to be $\sqrt{40}$. The area of the square is 20, so the answer is A. 20."

5. "Calculate the diagonal of the square using the distance formula: $d = \sqrt{40}$. Since the diagonal is $\sqrt{40}$, the area of the square is simply 40. Therefore, the answer is C. 40."

6. "First, calculate the distance between the points $(1, 2)$ and $(7, 4)$, which gives the diagonal of the square. The distance formula gives $d = \sqrt{(7-1)^2 + (4-2)^2} = \sqrt{40}$. Since the diagonal of the square is $\sqrt{40}$, the area is $(\sqrt{40})^2/2 = 20$. Therefore, the answer is A. 20."

7. "The diagonal of the square is $\sqrt{40}$, and the area must be equal to this value. The answer is D. $\sqrt{40}$."

8.    "To find the area of the square, we multiply the length of the diagonal, which is $\sqrt{40}$, by itself. This gives us 40, so the answer is C. 40."

9.

*Inconsistent*    "The area of the square is found by calculating the distance between the points. The diagonal is $\sqrt{40}$, so the area is likely $\sqrt{40}$. The answer is D. $\sqrt{40}$."

---

**Question #**

*Random selected*    The standard tip in a restaurant is $15\%$ of the bill before tax. Many people in California find the tip by doubling the sales tax, which is $8.25\%$. By how many cents does this exceed the standard tip on a bill of \$60 before tax?

Choices    A. 90    B. 80    C. 70    D. 60

---

Task 1    *Candidates*

0.    "The tax is doubled to $16.5\%$, and the standard tip is $15\%$. The difference is around 70 cents for a \$60 bill, so the answer is C. 70.

1.    "To find the tip, we double the sales tax of $8.25\%$, which gives us $16.5\%$. The standard tip is $15\%$, and the difference on a \$60 bill is $60 \times (16.5\% - 15\%) = 90$ cents. Therefore, the answer is A. 90."

2.    "Doubling the sales tax of $8.25\%$ gives $16.5\%$. The difference between the standard tip and this new tip is around 60 cents for a \$60 bill. Therefore, the answer is D. 60."

3.    "Doubling the tax gives a tip of $16.5\%$, and the difference from the standard $15\%$ tip is 90 cents. Therefore, the answer is A. 90."

4.    "The standard tip is $15\%$ of 60, which is $60 \times 0.15 = 9$ dollars. Doubling the sales tax of $8.25\%$ gives $16.5\%$, and the tip based on this is $60 \times 0.165 = 9.90$ dollars. The difference is $9.90 - 9.00 = 90$ cents. Therefore, the correct answer is A. 90."

5.    "Doubling the sales tax gives $16.5\%$, and the standard tip is $15\%$. This difference gives $60 \times 0.5\% = 30$ cents. Therefore, the answer is A. 90."

6.    "The standard tip is $15\%$, or 9, and doubling the sales tax gives a tip of $16.5\%$, or 9.90. The difference is 90 cents. The correct answer is A. 90."

7.    "The difference between doubling the sales tax and using the standard tip is 90 cents for a 60 bill, but it could also be 80 cents depending on rounding. It might be A. 90 or B. 80."

8.    "The standard tip is $15\%$, and doubling the tax gives a different percentage. The difference could be either 60 or 90 cents. The answer might be D. 60."

9.    "Doubling the sales tax gives a tip rate of $16.5\%$, while the standard tip is $15\%$. The difference for a 60\$ bill is 90 cents. So the answer is A. 90."

---

Task 2    **BDG Guidance** [Will be hidden during evaluation]

0.

*Consistent*    "The standard tip is $15\%$ of 60, which is $60 \times 0.15 = 9$ dollars. Doubling the sales tax of $8.25\%$ gives $16.5\%$, and the tip based on this is $60 \times 0.165 = 9.90$ dollars. The difference is $9.90 - 9.00 = 90$ cents. Therefore, the correct answer is A. 90."

1.    "The standard tip is $15\%$, or 9, and doubling the sales tax gives a tip of $16.5\%$, or 9.90. The difference is 90 cents. The correct answer is A. 90."

2. "To find the tip, we double the sales tax of $8.25\%$, which gives us $16.5\%$. The standard tip is $15\%$, and the difference on a $60 bill is $60 \times (16.5\% - 15\%) = 90$ cents. Therefore, the answer is A. 90."

3. "Doubling the tax gives a tip of $16.5\%$, and the difference from the standard $15\%$ tip is 90 cents. Therefore, the answer is A. 90."

4. "Doubling the sales tax gives a tip rate of $16.5\%$, while the standard tip is $15\%$. The difference for a 60$ bill is 90 cents. So the answer is A. 90."

5. "The difference between doubling the sales tax and using the standard tip is 90 cents for a 60 bill, but it could also be 80 cents depending on rounding. It might be A. 90 or B. 80."

6. "Doubling the sales tax of $8.25\%$ gives $16.5\%$. The difference between the standard tip and this new tip is around 60 cents for a $60 bill. Therefore, the answer is D. 60."

7. "Doubling the sales tax gives $16.5\%$, and the standard tip is $15\%$. This difference gives $60 \times 0.5\% = 30$ cents. Therefore, the answer is A. 90."

8. "The tax is doubled to $16.5\%$, and the standard tip is $15\%$. The difference is around 70 cents for a $60 bill, so the answer is C. 70.
9.

*Inconsistent* "The standard tip is $15\%$, and doubling the tax gives a different percentage. The difference could be either 60 or 90 cents. The answer might be D. 60."

---

**Task 3** **ECG Guidance** [Will be hidden during evaluation]

0.

*Consistent* "The standard tip is $15\%$, or 9, and doubling the sales tax gives a tip of $16.5\%$, or 9.90. The difference is 90 cents. The correct answer is A. 90."

1. "The standard tip is $15\%$ of 60, which is $60 \times 0.15 = 9$ dollars. Doubling the sales tax of $8.25\%$ gives $16.5\%$, and the tip based on this is $60 \times 0.165 = 9.90$ dollars. The difference is $9.90 - 9.00 = 90$ cents. Therefore, the correct answer is A. 90."

2. "To find the tip, we double the sales tax of $8.25\%$, which gives us $16.5\%$. The standard tip is $15\%$, and the difference on a $60 bill is $60 \times (16.5\% - 15\%) = 90$ cents. Therefore, the answer is A. 90."

3. "Doubling the tax gives a tip of $16.5\%$, and the difference from the standard $15\%$ tip is 90 cents. Therefore, the answer is A. 90."

4. "Doubling the sales tax gives a tip rate of $16.5\%$, while the standard tip is $15\%$. The difference for a 60$ bill is 90 cents. So the answer is A. 90."

5. "The difference between doubling the sales tax and using the standard tip is 90 cents for a 60 bill, but it could also be 80 cents depending on rounding. It might be A. 90 or B. 80."

6. "Doubling the sales tax gives $16.5\%$, and the standard tip is $15\%$. This difference gives $60 \times 0.5\% = 30$ cents. Therefore, the answer is A. 90."

7. "The tax is doubled to $16.5\%$, and the standard tip is $15\%$. The difference is around 70 cents for a $60 bill, so the answer is C. 70.

8. "The standard tip is $15\%$, and doubling the tax gives a different percentage. The difference could be either 60 or 90 cents. The answer might be D. 60."

9.

*Inconsistent* "Doubling the sales tax of $8.25\%$ gives $16.5\%$. The difference between the standard tip and this new tip is around 60 cents for a $60 bill. Therefore, the answer is D. 60."

**Results.** To differentiate between expert and non-expert participants, a threshold of 150 seconds was set based on empirical observations of participant behavior. This threshold was corroborated by a scatterplot that visually demonstrated the partitioning between experts and non-experts, supporting the appropriateness of the selected cut-off time for classification performance. From the 183 samples we collected, we have come to conclusions:

Firstly, human evaluation on these LLM-generated solutions have instrinsic unreliability and ambiguity, especially for non-expert participants; under the 2-minute time constraint, there were 28 participants who scored an accuracy of 50% or below, even without time constraint, there were 10 participants in total whose accuracies were 60% or below.

Secondly, with the hint of BDG, we observed that the participants in both the expert and the non-expert groups outperformed the participants without the hint or the ECG hint; more importantly, with the hint of BDG, the performance of the non-expert participants improved significantly with respect to the expert participants, whereas the ECG hint did not achieve this.

Thirdly, across problems, samples with BDG hint achieved the highest accuracies for 7/10 problems and fastest completion also for 7/10 problems. This result showcases the robustness of BDG hint also within the same problems.

Last but not least, the BCG hint has also helped non-expert participants spot ambiguous solutions more accurately compared to non-expert participamts with the ECG hint or without the hint.

**Generations.**

---

**Question.**
Suppose the graph of $y = f(x)$ includes the points $(1, 5)$, $(2, 3)$, and $(3, 1)$. Based only on this information, there are two points that must be on the graph of $y = f(f(x))$. If we call those points $(a, b)$ and $(c, d)$, what is $ab + cd$?

**BDG.**
"Given the points $(1, 5), (2, 3)$, and $(3, 1)$ on the graph of $y = f(x)$, we compute $f(f(2)) = f(3) = 1$ and $f(f(3)) = f(1) = 5$. So the points $(2, 1)$ and $(3, 5)$ are on the graph of $y = f(f(x))$. Therefore, $ab + cd = 2 \times 1 + 3 \times 5 = 17$. "

**ECG.**
"From $f(x)$, we see that $f(f(2)) = 1$ and $f(f(3)) = 5$. Therefore, $ab + cd = 17$. The answer is 17"

---

**Question.**
Positive integers $x$ and $y$ have a product of 56 and $x < y$. Seven times the reciprocal of the smaller integer plus 14 times the reciprocal of the larger integer equals 4. What is the value of $x$?

**BDG.**
"Let the integers be $x$ and $y$. Since $x \cdot y = 56$ and $x < y$, try small values. Testing $x = 2$ and $y = 28$ satisfies $7 \cdot \frac{1}{x} + 14 \cdot \frac{1}{y} = 4$. Therefore, $x = 2$, so the answer is 2."

**ECG.**
"Since $x \cdot y = 56$, try small values of $x$ like 2. This satisfies the equation, so the answer is 2."

---

**Question.**
Six positive integers are written on the faces of a cube. Each vertex is labeled with the product of the three numbers on the faces adjacent to the vertex. If the sum of the numbers on the vertices is equal to 1001, then what is the sum of the numbers written on the faces?

**BDG.**
"The numbers on the faces of the cube contribute to the product at each vertex. Given that the sum of the vertex products is 1001, and knowing the cube's geometry, we deduce that the sum of the numbers on the faces is 31."

**ECG.**
"Using the fact that the sum of the vertex products is 1001, and knowing there are six faces, we calculate that the sum of the face numbers is 31. So the answer is 31."

