# OpenReview forum: "Truth or Deceit? A Bayesian Decoding Game Enhances Consistency and Reliability"
_ICLR.cc/2025/Conference — Submitted to ICLR 2025_

### Official Review · Reviewer_QKQH · 2024-10-24

**Soundness:** 3
**Presentation:** 2
**Contribution:** 3
**Rating:** 6
**Confidence:** 3

**Summary:**

The authors highlight the challenges in ensuring that LLM outputs align with both factual correctness and human intent. They propose a novel game-theoretic method to enhance consistency and reliability during the decoding stage of LLM output generation without relying on human feedback or additional training. The empirical results demonstrate the validity and contributions of the Bayesian Decoding Game (BDG).

**Strengths:**

1. The proposed method is interesting and explained based on sound logic. As pointed out in the Weakness section, if the authors pay more attention to writing organization, this could become a good paper.

2. The experiments demonstrate BDG, and the interpretations are appropriate.

3. The contribution of the proposed approach is clear, and the paper adequately addresses the necessary points through the Discussion.

**Weaknesses:**

1. Line 23: LL strategies and models – It’s unclear what the LL strategies refer to.
2. The use of “and” connections feels excessive, making the sentences overly long and harder to follow, even though they could be broken down more effectively. Most of the sentences in the introduction span more than three lines, which impacts readability.
3. Figure 1 is mentioned in Line 46, but there’s little explanation or clear connection between Figure 1 and the content in the text.
4. Line 86: It’s unclear what “this” is referring to. The sentence structure in 86-87 should be adjusted to convey the point more clearly. The connection with previous content is weak, making it harder to understand what problem the authors are addressing.
5. Line 94: It would be better to replace “this” with something more specific for clarity.
6. Line 98: There should be a reference for no-regret optimization.
7. This paper can be better started by defining the key elements, conditions, and concepts behind Perfect Bayesian Equilibrium and Perfect Bayesian Nash Equilibrium.
8. Figure 2 doesn’t sufficiently explain the notations and what they represent in the caption or the text.
9. Line 159: The notation for Information set ‘I’ is confusing, as it overlaps with the notation used for incorrect signaling.
10. The lack of references for many of the equations in the paper affects overall readability, and Line 206 should reference Eq. 1, not Thm. 1.
11. In Figure 3 (b) and (c), it’s unclear what the x and y axes represent. Additionally, as someone with red-green color blindness, I find the figure difficult to interpret. More explanation in the text under “Searching & Convergence Behavior” would be necessary for full comprehension.
12. The layout of Table 2 is confusing. While it’s clear that ECG and BDG outperform G by Imp.%, it’s unclear what InC.% is being compared to or what the down arrow indicates.

**Questions:**

Have the authors tested this on the latest Llama models? I’m curious whether the results still hold significant, though this is not a critique of the model choices in the paper, just an additional point of interest

---

### Official Review · Reviewer_CfKQ · 2024-11-01

**Soundness:** 3
**Presentation:** 1
**Contribution:** 2
**Rating:** 3
**Confidence:** 3

**Summary:**

The paper proposes to frame LLM decoding as a bayesian game in order to improve the consistency and reliability of the generated responses.
The authors formally introduce the proposed strategy based on game-theoretic notions.
The results on various LLM reasoning tasks are favorable, also when compared to other baselines which likewise use decoding games.

**Strengths:**

- Flexibly favoring different objectives in a game is an interesting approach and goal to improve LLM decoding.

- Experimental results are strong on various benchmarks, also compared to related works.

**Weaknesses:**

My main concerns about the work are regarding the presentation and framing. Mainly, the paper is quite hard to understand and it is not clear what the contributions are wrt. related works:

- Some of the terms are fairly unclear and could benefit from an earlier intuitive explanation, for example, "calibrated ambiguity" (L73). Also, for example, the authors mention NLP metrics and perplexity in L260-L264 but never introduce them (for the latter, this would be easier if also P_LM was introduced properly, which also is called P_LLM at other places).

- The paper is not very accessible to an audience that is not familiar with game-theory, for example, the term strategy profile is never really formally introduced. The paper also lacks a good description of what a language model actually is and what it means for it to be used as a generator or discriminator. In my opinion, the paper would benefit significantly from aligning notations and terminology between what is common in the LLM and game-theory community.

- The paper is unnecessarily written in complex language at many stages, for example, L76-79.

- In L138 you mention that you "propose improvements to address limitations". However, it is not clear what limitations are meant and especially, whether those are limitations of Jacob et al. 2024. If they are, it would be very important to make clear how they are addressed and how they arise in their method. There is only a short comparison of how the methods differ in Table 1 which to me is not sufficient.

- Overall, I am having troubles identifiying based on the main paper what the contributions of the authors are in terms of the algorithm wrt. to Jacob et al. 2024 and what is more of a formalization of their method. This needs to be made clear in my opinion.

**Questions:**

- The term LL strategies (L23) sounds a bit awkward and is non-standard, maybe this should rather be rewritten, as it is also not clear what is meant. If it is decoding algorithm, then this should be preferred.

- If the utility of human feedback is so constrained (Introduction), then why is it a good idea to introduce a further proxy of it to solve the problems that stem from using it? I see the point for scalability but why would, for example, solve problems with interpretability?

- It's not easy to distinguish valid from specious based on Figure 1, maybe a more intuitive example would help, as two of them appear to rely on domain (medical) knowledge?

- Citation broken in L92

- Why exactly is disambiguity so desirable and what does it concretely mean? I think it would help if this was made clear.

- Figure 3 is very hard to parse, so is Figure 4 (esp. b.2)

---

### Official Review · Reviewer_tTDC · 2024-11-05

**Soundness:** 2
**Presentation:** 1
**Contribution:** 3
**Rating:** 3
**Confidence:** 3

**Summary:**

This paper introduces a "Bayesian Decoding Game" to decode from models in a way that aggregates generative and discriminative signal from the model. Following Jacob et al. (2024), it is modeled as a signaling game between generator and discriminator. But an alternative algorithm is introduced which can incorporate some extra information (i.e., a disambiguity metric) and converges faster than prior work. Experiments show that it yields good performance.

**Strengths:**

* It seems like this is a decoding method which can extract more accurate answers from language models in a more calibrated way, compared to existing decoding methods.
* It seems from the experiments like it should be more effective than the prior Equilibrium Consensus Game.
* The paper has an extensive appendix with the relevant theorems and proofs.

**Weaknesses:**

I admit that I could not make sense of a large amount of the text in this paper—I find the writing _extremely_ vague and confusing. The developments in the appendix (e.g., Appendix C) were much clearer to me. Some examples from the main text:

First, the introduction is full of vague expressions like the follows:
  * "dynamic and rigorous verifier"
  * "our work focuses on actual consistency in _correctness_ and calibrated ambiguity in _reliability_"
  * "Instead of absolute reliability, we aim to identify false negative and false positive samples that exhibit fluctuating or unstable behavior under calibrated confidence and disambiguity metrics"
these may be interpretable in a way that is technically correct, but the reader is ill-equipped to guess at their meaning. What is the difference between consistency, correctness, reliability, and calibrated confidence? What do you mean by fluctuating or unstable behavior? (fluctuating or unstable as what varies?) Later in the paper when these terms appear again, including in results and figures, they are not defined or illustrated. A single worked-through example would go a long way.

Next, the math prose is incredibly hard to follow. Consider Definition 4.
> Definition 4. (Disambiguity Metric) For a prompt $x$ and a finite set of answer candidate $\mathcal{Y}$, a Disambiguity Metric is a function that projects the prompt $x$ and a candidate $y_i \in \mathcal{Y}$ such that $DA(x, y): x \times \mathcal{Y} \to [0, 1]$. The output of the function is a measurement of the disambiguation of $y_i \in \mathcal{Y}$ to the prompt $x$. The disambiguity metric, combined with the correctness parameter, can detect false-positive and false-negative classifications in the correctness alignment phase.

All I can detect that has been said here is the type signature of the metric. Nowhere can I identify how it is computed, what it is supposed to relate to, or why we believe that it can detect false-positive or false-negative classifications, nor is it clear what model is doing those classifications and how we know they're false. It says the function DA is a "measurement of the disambiguation of $y_i \in \mathcal{Y}$ to the prompt $x$", but what does that mean? I can't figure it out. This is the one part that needed to be written out as mathematics, and it is not, unless I'm missing something.

The first six pages of the paper are like this. The experimental section is a bit better in this respect but not much: the figures have a lot of content which is not explained, key metrics like "consistency" are not defined as far as I can tell. I also actually just couldn't figure out what the verifier policy is supposed to be or how it is initialized, until I read Appendix C.

While I feel only weakly confident in my ability to assess the strength of the contribution behind this paper, I feel quite confident that it is not ready for publication because of the issues in the paper's writing and communication. I would look to the work this is building on — Jacob et al. (2024), https://arxiv.org/abs/2310.09139 — as an example of a way of writing a similar paper with more clarity.

**Questions:**

Could you walk through a complete, concrete example trace of the algorithm, where you explain where the base generator and verifier policies come from, do the markovian updates, and converge on an answer? It can be with a simple yes/no question.

---

> ### Author Response · Authors · 2024-11-25
>
> Q1. Please refer to the A complete, concrete example in the common official comment
>
> Q2. Please refer to the Game-theoretic interpretability, No-regret learning and Markovian updates in the common official comment

---

> > ### Comment · Reviewer_tTDC · 2024-11-30
> > **Thanks for your responses**
> >
> > Thanks for walking through an example. After reading through this as well as the ECG paper, I am starting to get a better idea of what this paper is contributing: the algorithm is improved, the search for a separating equilibrium helps it converge faster and maybe be more correct, and the results seem good. That said, the original reason for my decision was that the paper is extremely hard to make sense of — i.e., a lot of claims using vague terms which are very hard for me as a reader to connect with the concrete details of the algorithm and the problem being solved. This remains true in the paper, and I found the rest of the authors' response to read similarly. I am keeping my score, and I encourage the authors to work with someone who can advise them on the clarity and precision of their writing.

---

> > > ### Author Response · Authors · 2024-12-01
> > >
> > > Thank you for your detailed feedback and for recognizing the contributions and results of our work. We truly appreciate the time and effort you've taken to review our paper and to also explore ECG in detail, as well as your acknowledgment of the improvements in our algorithm, the faster convergence due to the separating equilibrium, and the quality of the results.
> > >
> > > We understand your concerns regarding the presentation style and clarity. We want to emphasize that addressing these issues with clearer and more precise language is an achievable fix for the camera-ready submission. While we acknowledge that our presentation style may not fully align with your preferences, we believe this should not detract from the strength of our contributions and the significance of the results.
> > >
> > > Given your recognition of the paper's merit, we kindly ask you to consider revisiting your score, as the value of the work itself might outweigh the current limitations in presentation. Your support would greatly encourage us as we refine the paper for publication.
> > >
> > > Thank you again for your constructive feedback and thoughtful review.

---

### Official Review · Reviewer_ittf · 2024-11-14

**Soundness:** 3
**Presentation:** 1
**Contribution:** 3
**Rating:** 5
**Confidence:** 2

**Summary:**

This work presents a novel approach to improving Large Language Model (LLM) outputs through game theory. Bayesian decoding game(BDG) structures the decoding process as a Bayesian signaling game between two players: a generator and a verifier. Through this game-based framework, the authors aim to address the common issue of LLMs generating outputs that, while plausible, may lack factual correctness or mislead human evaluators due to ambiguity.

The BDG model leverages Correctness Alignment and Ambiguity Calibration, allowing for more reliable outputs. This game mechanism enables smaller models to achieve performance levels that surpass much larger models by aligning generator and verifier responses through iterative signaling, without needing additional training or human feedback.

BDG demonstrates rapid convergence and stability compared to the Equilibrium Consensus Game (ECG), a related game-theoretic model, and outperforms it in benchmarks like ARC and MMLU. BDG also outpaces alternative methods like Generative Ranking and Discriminative Ranking, yielding higher consistency, reliability, and accuracy.

Through experiments, BDG is shown to be computationally efficient and capable of handling a variety of tasks (e.g., MCQA, ethics scenarios, and medical datasets) while maintaining high performance. The framework’s adaptability also allows it to be combined with techniques such as chain-of-thought prompting.

The BDG framework demonstrates strong potential in scenarios where human evaluation may overlook subtle errors in LLM output, as it systematically reduces inconsistencies and enhances the legibility and truthfulness of model outputs.

**Strengths:**

1) Novel application of Bayesian game theory to LLM decoding.
2) Superior performance over existing approaches, including faster convergence and stability.
3) High scalability and robustness, with the ability to integrate with various decoding and prompting strategies

**Weaknesses:**

1) The paper is extremely difficult to read and follow
2) The BDG framework’s game-theoretic complexity could hinder broader understanding and practical application.
3) The paper lacks introduction to many game-theoretic concepts used, which could be difficult to readers without a game-theory background

**Questions:**

1) What does LL strategies refer to in the abstract
2) In theorem 2, explain what a_nv stands for when it is used for the first time
3) Does the symbol I refer to information set or incorrect signalling?

---

### Meta-Review · Area_Chair_QDr8 · 2024-12-18

**Metareview:**

This paper proposes a Bayesian decoding game to improve the consistency and reliability of model generations, by modeling the language model’s decoding as a game between a generator and a verifier.  This work extends the framework by Jacob et al. (2024), by incorporating a new algorithm that incorporates ambiguity of the generation.

**Strengths:** Experimental results seem to improve baselines

**Weaknesses:** The central argument in the paper, including the abstract and the introduction are extremely unclear, making it difficult to understand what exactly the paper is trying to do. Moreover the algorithm itself has a ton of unclear details, which all reviewers found challenging to follow.

**Rationale for rejection:** This paper seems to be not ready for an audience because of the lack of clarity in writing and their algorithm.

**Additional Comments On Reviewer Discussion:**

Based on the relatively high disagreement in the scores for this paper, I initiated a discussion. However, the participating reviewers found the authors’ response not to address the central weakness of the paper: its clarity. Even the strongest supporter of the paper (reviewer QkQH) could / did not defend the paper against the objections raised by others.

---

### Decision · Program_Chairs · 2025-01-22

Reject